# WASSERSTEIN-2 GENERATIVE NETWORKS

**Alexander Korotin**
Skolkovo Institute of Science and Technology
*Moscow, Russia*
a.korotin@skoltech.ru

**Vage Egiazarian**
Skolkovo Institute of Science and Technology
*Moscow, Russia*
vage.egiazarian@skoltech.ru

**Arip Asadulaev**
ITMO University
*Saint Petersburg, Russia*
aripasadulaev@itmo.ru

**Aleksandr Safin**
Skolkovo Institute of Science and Technology
*Moscow, Russia*
aleksandr.safin@skoltech.ru

**Evgeny Burnaev**
Skolkovo Institute of Science and Technology
*Moscow, Russia*
e.burnaev@skoltech.ru

## ABSTRACT

We propose a novel end-to-end non-minimax algorithm for training optimal transport mappings for the quadratic cost (Wasserstein-2 distance). The algorithm uses input convex neural networks and a cycle-consistency regularization to approximate Wasserstein-2 distance. In contrast to popular entropic and quadratic regularizers, cycle-consistency does not introduce bias and scales well to high dimensions. From the theoretical side, we estimate the properties of the generative mapping fitted by our algorithm. From the practical side, we evaluate our algorithm on a wide range of tasks: image-to-image color transfer, latent space optimal transport, image-to-image style transfer, and domain adaptation.

## 1 INTRODUCTION

Generative learning framework has become widespread over the last couple of years tentatively starting with the introduction of generative adversarial networks (GANs) by Goodfellow et al. (2014). The framework aims to define a stochastic procedure to sample from a given complex probability distribution $\mathbb{Q}$ on a space $Y \subset \mathbb{R}^D$, e.g. a space of images. The usual generative pipeline includes sampling from tractable distribution $\mathbb{P}$ on space $\mathcal{X}$ and applying a generative mapping $g : \mathcal{X} \to \mathcal{Y}$ that transforms $\mathbb{P}$ into the desired $\mathbb{Q}$.

In many cases for probability distributions $\mathbb{P}, \mathbb{Q}$, there may exist several different generative mappings. For example, the mapping in Figure 1b seems to be better than the one in Figure 1a and should be preferred: the mapping in Figure 1b is straightforward, well-structured and invertible.

Existing generative learning approaches mainly do not focus on the structural properties of the generative mapping. For example, GAN-based approaches, such as $f$-GAN by Nowozin et al. (2016); Yadav et al. (2017), W-GAN by Arjovsky et al. (2017) and others Li et al. (2017); Mroueh & Sercu (2017), approximate generative mapping by a neural network with a problem-specific architecture.

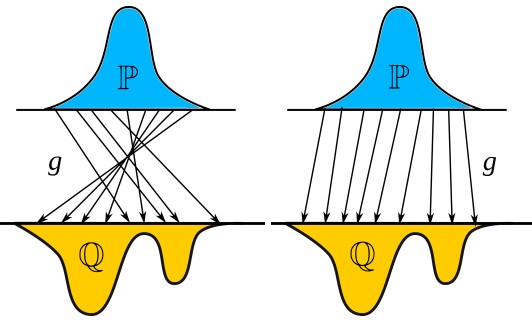

(a) An Arbitrary Mapping. (b) The Monotone Mapping.

Figure 1: Two possible generative mappings that transform distribution $\mathbb{P}$ to distribution $\mathbb{Q}$.

The reasonable question is how to find a generative mapping $g \circ \mathbb{P} = \mathbb{Q}$ that is **well-structured**. Typically, the better the structure of the mapping is, the easier it is to find such a mapping. There are many ways to define what the well-structured mapping is. But usually, such a mapping is expected to be continuous and, if possible, invertible. One may note that when $\mathbb{P}$ and $\mathbb{Q}$ are both one-dimensional ($\mathcal{X}, \mathcal{Y} \subset \mathbb{R}^1$), the only class of mappings $g : \mathcal{X} \to \mathcal{Y}$ satisfying these properties are monotone mappings[1], i.e. $\forall x, x' \in \mathcal{X}$ $(x \neq x')$ satisfying $\big(g(x) - g(x')\big) \cdot \big(x - x'\big) > 0$. The intuition of 1-dimensional spaces can be easily extended to $\mathcal{X}, \mathcal{Y} \subset \mathbb{R}^D$. We can require the similar condition to hold true: $\forall x, x' \in \mathcal{X}$ $(x \neq x')$

$$\langle g(x) - g(x'), x - x' \rangle > 0. \tag{1}$$

The condition (1) is called monotonicity, and every surjective function satisfying this condition is invertible. In one-dimensional case, for any pair of continuous $\mathbb{P}, \mathbb{Q}$ with non-zero density there exists a unique monotone generative map given by $g(x) = F_{\mathbb{Q}}^{-1}\big(F_{\mathbb{P}}(x)\big)$ McCann et al. (1995), where $F_{(\cdot)}$ is the cumulative distribution function of $\mathbb{P}$ or $\mathbb{Q}$. However, for $D > 1$ there might exist more than one generative monotone mapping. For example, when $\mathbb{P} = \mathbb{Q}$ are standard 2-dimensional Gaussian distributions, all rotations by angles $-\frac{\pi}{2} < \alpha < \frac{\pi}{2}$ are monotone and preserve the distribution.

One may impose uniqueness by considering only maximal Peyré (2018) monotone mappings $g : \mathcal{X} \to \mathcal{Y}$ satisfying $\forall N = 2, 3 \ldots$ and $N$ distinct points $x_1, \ldots, x_N \in \mathcal{X}$ ($N + 1 \equiv 1$):

$$\sum_{n=1}^{N} \langle g(x_n), x_n - x_{n+1} \rangle > 0. \tag{2}$$

The condition (2) is called **cycle monotonicity** and also implies "usual" monotonicity (1).

Importantly, for almost every two continuous probability distributions $\mathbb{P}, \mathbb{Q}$ on $\mathcal{X} = \mathcal{Y} = \mathbb{R}^D$ there exists a unique cycle monotone mapping $g : \mathcal{X} \to \mathcal{Y}$ satisfying $g \circ \mathbb{P} = \mathbb{Q}$, see McCann et al. (1995). Thus, instead of searching for arbitrary generative mapping, one may significantly reduce the considered approximating class of mappings by using only cycle monotone ones.

According to Rockafellar (1966), every cycle monotone mapping $g$ is contained in a sub-gradient of some convex function $\psi : \mathcal{X} \to \mathbb{R}$. Thus, every convex class of functions may produce cycle monotone mappings (by considering sub-gradients of these functions). In practice, deep **input convex neural networks** (ICNNs, see Amos et al. (2017)) can be used as a class of convex functions.

Formally, to fit a cycle monotone generative mapping, one may apply any existing approach, such as GANs Goodfellow et al. (2014), with the set of generators restricted to gradients of ICNN. However, GANs typically require solving a minimax optimization problem.

It turns out that the cycle monotone generators are strongly related to **Wasserstein-2** distance ($\mathbb{W}_2$). The approaches by Taghvaei & Jalali (2019); Makkuva et al. (2019) use dual form of $\mathbb{W}_2$ to find the **optimal** generative mapping which is cycle monotone. The predecessor of both approaches is the gradient-descent algorithm for computing $\mathbb{W}_2$ distance by Chartrand et al. (2009). The drawback of all these methods is similar to the one of GANs – their optimization objectives are minimax.

Cyclically monotone generators require that both spaces $\mathcal{X}$ and $\mathcal{Y}$ have the same dimension, which poses no practical limitation. Indeed, it is possible to combine a generative mapping with a decoder of a pre-trained autoencoder, i.e. train a generative mapping into a latent space. It should be also noted that the cases with equal dimensions of $\mathcal{X}$ and $\mathcal{Y}$ are common in computer vision. The typical example is image-to-image style transfer when both the input and the output images have the same size and number of channels. Other examples include image-to-image color transfer, domain adaptation, etc.

In this paper, we develop the concept of cyclically monotone generative learning. The **main contributions** of the paper are as follows:

1. Developing an end-to-end non-minimax algorithm for training cyclically monotone generative maps, i.e. optimal maps for quadratic transport cost (Wasserstein-2 distance).
2. Proving theoretical bound on the approximation properties of the transport mapping fitted by the developed approach.
3. Developing a class of Input Convex Neural Networks whose gradients are used to approximate cyclically monotone mappings.

---

[1]We consider only monotone **increasing** mappings. Decreasing mappings have analogous properties.

4. Demonstrating the performance of the method in practical problems of image-to-image color transfer, mass transport in latent spaces, image-to-image style translation and domain adaptation.

Our algorithm extends the approach of Makkuva et al. (2019), eliminates minimax optimization imposing cyclic regularization and solves non-minimax optimization problem. At the result, the algorithm **scales well** to high dimensions and **converges** up to 10x times **faster** than its predecessors.

**The paper is structured as follows.** Section 2 is devoted to Related Work. In Section 3, we give the necessary mathematical tools on Wasserstein-2 optimal transport. In Section 4, we derive our algorithm and state our main theoretical results. In Section 5, we provide the results of computational experiments. In Appendix A, we prove our theoretical results. In Appendix B, we describe the particular architectures of ICNN that we use for experiments. In Appendix C, additional experiments and training details are provided.

## 2 RELATED WORK

Modern generative learning is mainly associated with **Generative Adversarial Networks** (GANs) Goodfellow et al. (2014); Arjovsky et al. (2017). Basic GAN model consists of two competing networks: generator $g$ and discriminator $d$. Generator $g$ takes as input samples $x$ from given distribution $\mathbb{P}$ and tries to produce realistic samples from real data distribution $\mathbb{Q}$. Discriminator $d$ attempts to distinguish between generated and real distributions $g \circ \mathbb{P}$ and $\mathbb{Q}$ respectively. Formally, it approximates a dissimilarity measure between $g \circ \mathbb{P}$ and $\mathbb{Q}$ (e.g. $f$-divergence Nowozin et al. (2016) or Wasserstein-1 distance Arjovsky et al. (2017)). Although superior performance is reported for many applications of GANs Karras et al. (2017); Mirza & Osindero (2014), training such models is always hard due to the minimax nature of the optimization objective.

Another important branch of generative learning is related to the theory of **Optimal Transport** (OT) Villani (2008); Peyré et al. (2019). OT methods seek generative mapping[2] $g : \mathcal{X} \to \mathcal{Y}$, optimal in the sense of the given transport cost $c : \mathcal{X} \times \mathcal{Y} \to \mathbb{R}$:

$$\text{Cost}(\mathbb{P}, \mathbb{Q}) = \min_{g \circ \mathbb{P} = \mathbb{Q}} \int_{\mathcal{X}} c\big(x, g(x)\big) d\mathbb{P}(x). \tag{3}$$

Equation (3) is also known as Monge's formulation of optimal transportation Villani (2008).

The principal OT generative method Seguy et al. (2017) is based on optimizing the **regularized dual form** of the transport cost (3). It fits two potentials $\psi, \overline{\psi}$ (primal and conjugate) and then uses the barycentric projection to establish the desired (third) generative network $g$. Although the method uses non-minimax optimization objective, it is not **end-to-end** (consists of two sequential steps).

In the case of quadratic transport cost $c(x, y) = \frac{\|x-y\|^2}{2}$, the value (3) is known as the square of **Wasserstein-2** distance:

$$\mathbb{W}_2^2(\mathbb{P}, \mathbb{Q}) = \min_{g \circ \mathbb{P} = \mathbb{Q}} \int_{\mathcal{X}} \frac{\|x - g(x)\|^2}{2} d\mathbb{P}(x). \tag{4}$$

It has been well studied in literature Brenier (1991); McCann et al. (1995); Villani (2003; 2008) and has many useful properties which we discuss in Section 3 in more detail. The optimal mapping for the quadratic cost is cyclically monotone. Several algorithms exist Lei et al. (2019); Taghvaei & Jalali (2019); Makkuva et al. (2019) for finding this mapping.

The recent approach by Taghvaei & Jalali (2019) uses the gradient-descent-based algorithm by Chartrand et al. (2009) for computing $\mathbb{W}_2$. The key idea is to approximate the optimal potential $\psi^*$ by an ICNN Amos et al. (2017), and extract the optimal generator $g^*$ from its gradient $\nabla \psi^*$. The method is impractical due to high computational complexity: during the main optimization cycle, it solves an additional optimization sub-problem. The inner problem is convex but computationally costly. This was noted in the original paper and de-facto confirmed by the lack of experiments with complex distributions. A refinement of this approach is proposed by Makkuva et al. (2019). The inner optimization sub-problem is removed, and a network is used to approximate its solution. This speeds up the computation, but the problem is still minimax.

---

[2]Commonly, in OT it is assumed that $\dim \mathcal{X} = \dim \mathcal{Y}$.

## 3 PRELIMINARIES

In the section, we recall the properties of $\mathbb{W}_2$ distance (4) and its relation to cycle monotone mappings.

Throughout the paper, we assume that $\mathbb{P}$ and $\mathbb{Q}$ are **continuous** distributions on $\mathcal{X} = \mathcal{Y} = \mathbb{R}^D$ with **finite second moments**.[3] This condition guarantees that (3) is well-defined in the sense that the optimal mapping $g^*$ always exists. It follows from (Villani, 2003, Brenier's Theorem 2.12) that its restriction to the support of $\mathbb{P}$ is unique (up to the values on the small sets) and invertible. The symmetric characteristics apply to its inverse $(g^*)^{-1}$, which induces symmetry to definition (4) for quadratic cost. According to Villani (2003), the dual form of (4) is given by

$$\mathbb{W}_2^2(\mathbb{P}, \mathbb{Q}) = \underbrace{\int_{\mathcal{X}} \frac{\|x\|^2}{2} d\mathbb{P}(x) + \int_{\mathcal{Y}} \frac{\|y\|^2}{2} d\mathbb{Q}(y)}_{\text{Const}(\mathbb{P}, \mathbb{Q})} - \underbrace{\min_{\psi \in \text{Convex}} \left[ \int_{\mathcal{X}} \psi(x) d\mathbb{P}(x) + \int_{\mathcal{Y}} \overline{\psi}(y) d\mathbb{Q}(y) \right]}_{\text{Corr}(\mathbb{P}, \mathbb{Q})}, \quad (5)$$

where the minimum is taken over all the convex functions (potentials) $\psi : \mathcal{X} \to \mathbb{R} \cup \{\infty\}$, and $\overline{\psi}(y) = \max_{x \in \mathcal{X}} \left( \langle x, y \rangle - \psi(x) \right)$ is the **convex conjugate** Fenchel (1949) to $\psi$, which is also a convex function, $\overline{\psi} : \mathcal{Y} \to \mathbb{R} \cup \{\infty\}$.

We call the value of the minimum in (5) **cyclically monotone correlations** and denote it by $\text{Corr}(\mathbb{P}, \mathbb{Q})$. By equating (5) with (4), one may derive the formula

$$\text{Corr}(\mathbb{P}, \mathbb{Q}) = \max_{g \circ \mathbb{P} = \mathbb{Q}} \int_{\mathcal{X}} \langle x, g(x) \rangle d\mathbb{P}(x). \quad (6)$$

Note that $\left( - \text{Corr}(\mathbb{P}, \mathbb{Q}) \right)$ can be viewed as an optimal transport cost for **bilinear cost function** $c(x, y) = -\langle x, y \rangle$, see McCann et al. (1995). Thus, searching for optimal transport map $g^*$ for $\mathbb{W}_2$ is equivalent to finding the mapping which maximizes correlations (6).

It is known for $\mathbb{W}_2$ distance that the gradient $g^* = \nabla \psi^*$ of optimal potential $\psi^*$ readily gives the minimizer of (4), see Villani (2003). Being a gradient of a convex function, it is necessarily cycle monotone. In particular, the inverse mapping can be obtained by taking the gradient w.r.t. input of the conjugate of optimal potential $\overline{\psi^*}(y)$ McCann et al. (1995). Thus, we have

$$(g^*)^{-1}(y) = \left( \nabla \psi^* \right)^{-1}(y) = \nabla \overline{\psi^*}(y). \quad (7)$$

In fact, one may approximate the primal potential $\psi$ by a parametric class $\Theta$ of input convex functions $\psi_\theta$ and optimize correlations

$$\min_{\theta \in \Theta} \text{Corr}(\mathbb{P}, \mathbb{Q} \mid \psi_\theta) = \min_{\theta \in \Theta} \left[ \int_{\mathcal{X}} \psi_\theta(x) d\mathbb{P}(x) + \int_{\mathcal{Y}} \overline{\psi_\theta}(y) d\mathbb{Q}(y) \right] \quad (8)$$

in order to extract the approximate optimal generator $g_{\theta^\dagger} : \mathcal{X} \to \mathcal{Y}$ from the approximate potential $\psi_{\theta^\dagger}$. Note that in general it is not true that $g_{\theta^\dagger} \circ \mathbb{P}$ will be equal to $\mathbb{Q}$. However, we prove that if $\text{Corr}(\mathbb{P}, \mathbb{Q} \mid \psi_{\theta^\dagger})$ is close to $\text{Corr}(\mathbb{P}, \mathbb{Q})$, then $g_{\theta^\dagger} \circ \mathbb{P} \approx \mathbb{Q}$, see our Theorem A.3 in Appendix A.2.

The optimization of (8) can be performed via stochastic gradient descent. It is possible to get rid of conjugate $\overline{\psi_\theta}$ and extract an analytic formula for the gradient of (8) w.r.t. parameters $\theta$ by using $\psi_\theta$ only, see the derivations in Taghvaei & Jalali (2019); Chartrand et al. (2009):

$$\frac{\partial \text{Corr}(\mathbb{P}, \mathbb{Q} \mid \psi_\theta)}{\partial \theta} = \int_{\mathcal{X}} \frac{\partial \psi_\theta(x)}{\partial \theta} d\mathbb{P}(x) - \int_{\mathcal{Y}} \frac{\partial \psi_\theta(\hat{x})}{\partial \theta} d\mathbb{Q}(y),$$

where $\frac{\partial \psi_\theta}{\partial \theta}$ in the second integral is computed at $\hat{x} = (\nabla \psi_\theta)^{-1}(y)$, i.e. inverse value of $y$ for $\nabla \psi_\theta$.

In practice, both integrals are replaced by their Monte Carlo estimates over random mini-batches from $\mathbb{P}$ and $\mathbb{Q}$. Yet to compute the second integral, one needs to recover the inverse values of the current mapping $\nabla \psi_\theta$ for all $y \sim \mathbb{Q}$ in the mini batch. To do this, the following optimization sub-problem has to be solved

$$\hat{x} = (\nabla \psi_\theta)^{-1}(y) \Leftrightarrow \hat{x} = \underset{x \in \mathcal{X}}{\arg\max} \left( \langle x, y \rangle - \psi_\theta(x) \right) \quad (9)$$

---

[3]In practice, the continuity condition can be assumed to hold true. Indeed, widely used heuristics, such as adding small Gaussian noise to data Sønderby et al. (2016), make considered distributions to be continuous.

for each $y \sim \mathbb{Q}$ in the mini batch. The optimization problem (9) is convex but **complex** because it requires computing the gradient of $\psi_\theta$ multiple times. It is computationally costly since $\psi_\theta$ is in general a large neural network. Besides, during iterations over $\theta$, each time a new independent batch of samples arrives. This makes it hard to use the information on the solution of (9) from the previous gradient descent step over $\theta$ in (8).

## 4 AN END-TO-END NON-MINIMAX ALGORITHM

In Subsection 4.1, we describe our novel end-to-end algorithm with non-minimax optimization objective for fitting cyclically monotone generative mappings. In Subsection 4.2, we state our main theoretical results on approximation properties of the proposed algorithm.

### 4.1 ALGORITHM

To simplify the inner optimization procedure for inverting the values of current $\nabla\psi_\theta$, one may consider the following **variational approximation** of the main objective:

$$
\min_{\psi \in \text{Convex}} \text{Corr}(\mathbb{P}, \mathbb{Q}|\psi) = \min_{\psi \in \text{Convex}} \left[ \int_{\mathcal{X}} \psi(x) d\mathbb{P}(x) + \int_{\mathcal{Y}} \overbrace{\max_{x \in \mathcal{X}} \left[ \langle x, y \rangle - \psi(x) \right]}^{= \overline{\psi}(y)} d\mathbb{Q}(y) \right] =
$$

$$
\min_{\psi \in \text{Convex}} \left[ \int_{\mathcal{X}} \psi(x) d\mathbb{P}(x) + \max_{T \in \mathcal{Y}^{\mathcal{X}}} \int_{\mathcal{Y}} \left[ \langle T(y), y \rangle - \psi(T(y)) \right] d\mathbb{Q}(y) \right], \quad (10)
$$

where by considering arbitrary measurable functions $T$, we obtain a **variational lower bound** which matches the entire value for $T = \left(\nabla\psi\right)^{-1}(y) = \nabla\overline{\psi}(y)$. Thus, a possible approach is to approximate both primal and dual potentials by two different networks $\psi_\theta$ and $\overline{\psi_\omega}$ and solve the optimization problem w.r.t. parameters $\theta, \omega$, e.g. by stochastic gradient descent/ascent Makkuva et al. (2019). Yet such a problem is still minimax. Thus, it suffers from typical problems such as convergence to local saddle points, instabilities during training and usually requires non-trivial hyperparameters choice.

We propose a method to get rid of the minimax objective by imposing additional regularization. Our **key idea** is to add regularization term $R_{\mathcal{Y}}(\theta, \omega)$ which stimulates **cycle consistency** Zhu et al. (2017), i.e. optimized generative mappings $g_\theta = \nabla\psi_\theta$ and $g_\omega^{-1} = \nabla\overline{\psi_\omega}$ should be mutually inverse:

$$
R_{\mathcal{Y}}(\theta, \omega) = \int_{\mathcal{Y}} \|g_\theta \circ g_\omega^{-1}(y) - y\|^2 d\mathbb{Q}(y) = \int_{\mathcal{Y}} \|\nabla\psi_\theta \circ \nabla\overline{\psi_\omega}(y) - y\|^2 d\mathbb{Q}(y). \quad (11)
$$

From the previous discussion and equation (7), we see that cycle consistency is a quite natural condition for $\mathbb{W}_2$ distance. More precisely, if $\nabla\psi_\theta$ and $\nabla\overline{\psi_\omega}$ are exactly inverse to each other (assuming $\nabla\psi_\theta$ is injective), then $\overline{\psi_\omega}$ is a convex conjugate to $\psi_\theta$ up to a constant.

In contrast to regularization used in Seguy et al. (2017), the proposed penalties use not the values of the potentials $\psi_\theta, \overline{\psi_\omega}$ itself but the values of their gradients (generators). This helps to stabilize the value of the regularization term which in the case of Seguy et al. (2017) may take extremely high values due to the fact that convex potentials grow fast in absolute value.[4]

Our proposed regularization leads to the following **non-minimax optimization objective** ($\lambda > 0$):

$$
\min_{\theta, \omega} \underbrace{\left[ \left( \int_{\mathcal{X}} \psi_\theta(x) d\mathbb{P}(x) + \int_{\mathcal{Y}} \left[ \langle \nabla\overline{\psi_\omega}(y), y \rangle - \psi_\theta(\nabla\overline{\psi_\omega}(y)) \right] d\mathbb{Q}(y) \right) + \frac{\lambda}{2} R_{\mathcal{Y}}(\theta, \omega) \right]}_{\text{Corr}(\mathbb{P}, \mathbb{Q}|\psi_\theta, \overline{\psi_\omega}; \lambda)}. \quad (12)
$$

The practical optimization procedure is given in Algorithm 1. We replace all the integrals by Monte Carlo estimates over random mini-batches from $\mathbb{P}$ and $\mathbb{Q}$. To perform optimization, we use the stochastic gradient descent over parameters $\theta, \omega$ of primal $\psi_\theta$ and dual $\overline{\psi_\omega}$ potentials.

We use the automatic differentiation to evaluate $\nabla\overline{\psi_\omega}$, $\nabla\psi_\theta$ and the gradients of (12) w.r.t. parameters $\theta, \omega$. The time required to compute the gradient of (12) w.r.t. $\theta, \omega$ is comparable by a constant factor

---

[4]For example, in the case of identity map $g(x) = \nabla\psi(x) = x$, we have quadratic growth: $\psi = \frac{\|x\|^2}{2} + c$.

---

**Algorithm 1:** Numerical Procedure for Optimizing Regularized Correlations (12)

---

**Input :** Distributions $\mathbb{P}, \mathbb{Q}$ with sample access; cycle-consistency regularizer coefficient $\lambda > 0$; a pair of input-convex neural networks $\psi_\theta$ and $\overline{\psi_\omega}$; batch size $K > 0$;

**for** $t = 1, 2, \ldots$ **do**

    1. Sample batches $X \sim \mathbb{P}$ and $Y \sim \mathbb{Q}$;

    2. Compute the Monte-Carlo estimate of the correlations:

$$\mathcal{L}_{\text{Corr}} = \frac{1}{K}\left[\sum_{x \in X} \psi_\theta(x) + \sum_{y \in Y}\left[\langle\nabla\overline{\psi_\omega}(y), y\rangle - \psi_\theta\big(\nabla\overline{\psi_\omega}(y)\big)\right]\right];$$

    3. Compute the Monte-Carlo estimate of the cycle-consistency regularizer:

$$\mathcal{L}_{\text{Cycle}} := \frac{1}{K}\sum_{y \in Y}\|\nabla\psi_\theta \circ \nabla\overline{\psi_\omega}(y) - y\|_2^2;$$

    4. Compute the total loss $\mathcal{L}_{\text{Total}} := \mathcal{L}_{\text{Corr}} + \frac{\lambda}{2} \cdot \mathcal{L}_{\text{Cycle}}$;

    5. Perform a gradient step over $\{\theta, \omega\}$ by using $\frac{\partial\mathcal{L}_{\text{Total}}}{\partial\{\theta,\omega\}}$;

**end**

---

to the time required to compute the value of $\psi_\theta(x)$. We empirically measured that this factor roughly equals 8-12, depending on the particular architecture of ICNN $\psi_\theta(x)$. We discuss the time complexity of a gradient step of our method in a more detail in Appendix C.2.

In Subsection 5.1, we show that our non-minimax approach **converges up to 10x times faster** than minimax alternatives by Makkuva et al. (2019) and Taghvaei & Jalali (2019).

## 4.2 APPROXIMATION PROPERTIES

Our gradient-descent-based approach described in Subsection 4.1 computes $\text{Corr}(\mathbb{P}, \mathbb{Q})$ by approximating it with a restricted sets of convex potentials. Let $(\psi^\dagger, \overline{\psi^\ddagger})$ be a pair of potentials obtained by the optimization of correlations. Formally, the fitted generators $g^\dagger = \nabla\psi^\dagger$ and $(g^\ddagger)^{-1} = \nabla\overline{\psi^\ddagger}$ are byproducts of optimization (12). We provide guarantees that the generated distribution $g^\dagger \circ \mathbb{P}$ is indeed close to $\mathbb{Q}$ as well as the inverse mapping $(g^\ddagger)^{-1}$ pushes $\mathbb{Q}$ close to $\mathbb{P}$.

**Theorem 4.1** (Generative Property for Approximators of Regularized Correlations). *Let $\mathbb{P}, \mathbb{Q}$ be two continuous probability distributions on $\mathcal{X} = \mathcal{Y} = \mathbb{R}^D$ with finite second moments. Let $\psi^* : \mathcal{X} \to \mathbb{R}$ be the optimal convex potential:*

$$\psi^* = \underset{\psi \in \text{Convex}}{\arg\min}\ \text{Corr}(\mathbb{P}, \mathbb{Q}|\psi) = \underset{\psi \in \text{Convex}}{\arg\min}\left[\int_\mathcal{X}\psi(x)d\mathbb{P}(x) + \int_\mathcal{Y}\overline{\psi}(y)d\mathbb{Q}(y)\right]. \quad (13)$$

*Let two differentiable convex functions $\psi^\dagger : \mathcal{X} \to \mathbb{R}$ and $\overline{\psi^\ddagger} : \mathcal{Y} \to \mathbb{R}$ satisfy for some $\epsilon \in \mathbb{R}$:*

$$\text{Corr}\big(\mathbb{P}, \mathbb{Q} \mid \psi^\dagger, \overline{\psi^\ddagger}; \lambda\big) \leq \left[\int_\mathcal{X}\psi^*(x)d\mathbb{P}(x) + \int_\mathcal{Y}\overline{\psi^*}(y)d\mathbb{Q}(y)\right] + \epsilon = \underbrace{\text{Corr}(\mathbb{P}, \mathbb{Q})}_{\text{Equals (6)}} + \epsilon. \quad (14)$$

*Assume that $\psi^\dagger$ is $\beta^\dagger$-strongly convex ($\beta^\dagger > \frac{1}{\lambda} > 0$) and $\mathcal{B}^\dagger$-smooth ($\mathcal{B}^\dagger \geq \beta^\dagger$). Assume that $\overline{\psi^\ddagger}$ has bijective gradient $\nabla\overline{\psi^\ddagger}$. Then the following inequalities hold true:*

1. **Correlation Upper Bound** *(regularized correlations dominate over the true ones)*

$$\text{Corr}\big(\mathbb{P}, \mathbb{Q} \mid \psi^\dagger, \overline{\psi^\ddagger}; \lambda\big) \geq \text{Corr}\big(\mathbb{P}, \mathbb{Q}\big) \qquad (i.e.\ \epsilon \geq 0);$$

2. **Forward Generative Property** *(mapping $g^\dagger = \nabla\psi^\dagger$ pushes $\mathbb{P}$ to be $O(\epsilon)$-close to $\mathbb{Q}$)*

$$\mathbb{W}_2^2(g^\dagger \circ \mathbb{P}, \mathbb{Q}) = \mathbb{W}_2^2(\nabla\psi^\dagger \circ \mathbb{P}, \mathbb{Q}) \leq \frac{(\mathcal{B}^\dagger)^2 \cdot \epsilon}{\lambda\beta^\dagger - 1} \cdot \left[\frac{1}{\sqrt{\beta^\dagger}} + \sqrt{\lambda}\right]^2;$$

3. **Inverse Generative Property** *(mapping $(g^\ddagger)^{-1} = \nabla\overline{\psi^\ddagger}$ pushes $\mathbb{Q}$ to be $O(\epsilon)$-close to $\mathbb{P}$)*

$$\mathbb{W}_2^2\big((g^\ddagger)^{-1} \circ \mathbb{Q}, \mathbb{P}\big) = \mathbb{W}_2^2(\nabla\overline{\psi^\ddagger} \circ \mathbb{Q}, \mathbb{P}) \leq \frac{\epsilon}{\beta^\dagger - \frac{1}{\lambda}}.$$

Informally, Theorem 4.1 states that the better we approximate correlations between $\mathbb{P}$ and $\mathbb{Q}$ by potentials $\psi^\dagger, \overline{\psi^\ddagger}$, the closer we expect generated distributions $g^\dagger \circ \mathbb{P}$ and $(g^\ddagger)^{-1} \circ \mathbb{Q}$ to be to $\mathbb{Q}$ and $\mathbb{P}$ respectively in the $\mathbb{W}_2$ sense. We prove Theorem 4.1 and provide extra discussion on smoothness and strong convexity in Section A.2. Additionally, we derive Theorem A.3 which states analogous generative properties for the mapping obtained by the base method (8) with single potential and no regularization.

Due to the Forward Generative property of Theorem 4.1, one may view the optimization of regularized correlations (12) as a process of minimizing $\mathbb{W}_2^2$ between the forward generated $g^\dagger \circ \mathbb{P}$ and true $\mathbb{Q}$ distributions (same applies to the inverse property). Wasserstein-2 distance prevents **mode dropping** for distant modes due to the quadratic cost. See the experiment in Figure 9 in Appendix C.3.

The following Theorem demonstrates that we actually can approximate correlations as well as required if the approximating classes of functions for potentials are large enough.

**Theorem 4.2** (Approximability of Correlations). *Let $\mathbb{P}, \mathbb{Q}$ be two continuous probability distributions on $\mathcal{X} = \mathcal{Y} = \mathbb{R}^D$ with finite second moments. Let $\psi^* : \mathcal{Y} \to \mathbb{R}$ be the optimal convex potential.*

*Let $\Psi_\mathcal{X}, \overline{\Psi}_\mathcal{Y}$ be classes of differentiable convex functions $\mathcal{X} \to \mathbb{R}$ and $\mathcal{Y} \to \mathbb{R}$ respectively and*

*1. $\exists \psi^\mathcal{X} \in \Psi_\mathcal{X}$ with $\epsilon_\mathcal{X}$-close gradient to the forward mapping $\nabla \psi^*$ in $\mathcal{L}^2(\mathcal{X} \to \mathbb{R}^D, \mathbb{P})$ sense:*

$$\|\nabla \psi^\mathcal{X} - \nabla \psi^*\|_\mathbb{P}^2 \overset{def}{=} \int_\mathcal{X} \|\nabla \psi^\mathcal{X}(y) - \nabla \psi^*(y)\|^2 d\mathbb{P}(y) \leq \epsilon_\mathcal{X},$$

*and $\psi^\mathcal{X}$ is $\mathcal{B}^\mathcal{X}$-smooth;*

*2. $\exists \overline{\psi^\mathcal{Y}} \in \overline{\Psi}_\mathcal{Y}$ with $\epsilon_\mathcal{Y}$-close gradient to the inverse mapping $\nabla \overline{\psi^*}$ in $\mathcal{L}^2(\mathcal{Y} \to \mathbb{R}^D, \mathbb{Q})$ sense:*

$$\|\nabla \overline{\psi^\mathcal{Y}} - \nabla \overline{\psi^*}\|_\mathbb{Q}^2 \overset{def}{=} \int_\mathcal{Y} \|\nabla \overline{\psi^\mathcal{Y}}(y) - \nabla \overline{\psi^*}(y)\|^2 d\mathbb{Q}(y) \leq \epsilon_\mathcal{Y}.$$

*Let $(\psi^\dagger, \overline{\psi^\ddagger})$ be the minimizers of the regularized correlations within $\Psi_\mathcal{X} \times \overline{\Psi}_\mathcal{Y}$:*

$$(\psi^\dagger, \overline{\psi^\ddagger}) = \underset{\psi \in \Psi_\mathcal{X}, \overline{\psi'} \in \overline{\Psi}_\mathcal{Y}}{\arg\min} \text{Corr}(\mathbb{P}, \mathbb{Q} \mid \psi, \overline{\psi'}; \lambda). \tag{15}$$

*Then the regularized correlations for $(\psi^\dagger, \overline{\psi^\ddagger})$ satisfy the following inequality:*

$$\text{Corr}(\mathbb{P}, \mathbb{Q} \mid \psi^\dagger, \overline{\psi^\ddagger}; \lambda) \leq \text{Corr}(\mathbb{P}, \mathbb{Q}) + \left[\frac{\lambda}{2}(\mathcal{B}^\mathcal{X}\sqrt{\epsilon_\mathcal{Y}} + \sqrt{\epsilon_\mathcal{X}})^2 + (\mathcal{B}^\mathcal{X}\sqrt{\epsilon_\mathcal{Y}} + \sqrt{\epsilon_\mathcal{X}}) \cdot (\sqrt{\epsilon_\mathcal{Y}}) + \frac{\mathcal{B}^\mathcal{X}}{2}\epsilon_\mathcal{Y}\right], \tag{16}$$

*i.e. regularized correlations do not exceed true correlations plus $O(\epsilon_\mathcal{X} + \epsilon_\mathcal{Y})$ term.*

By combining Theorem 4.2 with Theorem 4.1, we conclude that solutions $\psi^\dagger, \overline{\psi^\ddagger}$ of (15) push $\mathbb{P}$ and $\mathbb{Q}$ to be $O(\epsilon_\mathcal{X} + \epsilon_\mathcal{Y})$-close to $\mathbb{Q}$ and $\mathbb{P}$ respectively. In practice, it is reasonable to use input-convex neural networks as classes of functions $\Psi_\mathcal{X}, \overline{\Psi}_\mathcal{Y}$. Fully-connected ICNNs satisfy universal approximation property Chen et al. (2018).

In Appendix A.3, we prove that our method can be applied to the latent space scenario. Theorem A.4 states that the distance between the target and generated (latent space generative map combined with the decoder) distributions can upper bounded by the quality of the latent fit and the reconstruction loss of the auto-encoder. In Appendix A.4, we prove Theorem A.5, which demonstrates how our method can be applied to non-continuous distributions $\mathbb{P}$ and $\mathbb{Q}$.

## 5 EXPERIMENTS

In this section, we experimentally evaluate the proposed model.[5] In Subsection 5.1, we apply our method to estimate optimal transport maps in the Gaussian setting. In Subsection 5.2, we consider

---

[5]The code is written on **PyTorch** framework and is publicly available at

`https://github.com/iamalexkorotin/Wasserstein2GenerativeNetworks`.

latent space mass transport. In Subsection 5.3, we experiment with image-to-image style translation. In Appendix C, we provide training details and **additional experiments** on color transfer, domain adaptation and toy examples. The architectures of input convex networks that we use (Dense/Conv ICNNs) are described in Appendix B. The provided results are not intended to represent the state-of-the-art for any particular task – the goal is to show the feasibility of our approach and architectures.

## 5.1 OPTIMAL TRANSPORT BETWEEN GAUSSIANS

We test our method in the Gaussian setting $\mathbb{P}, \mathbb{Q} = \mathcal{N}(\mu_{\mathbb{P}}, \Sigma_{\mathbb{P}}), \mathcal{N}(\mu_{\mathbb{Q}}, \Sigma_{\mathbb{Q}})$. It is one of the few setups with the ground truth OT mapping existing in a closed form. We compare our method [W2GN] with quadratic regularization approach by Seguy et al. (2017) [LSOT] and minimax approaches by Taghvaei & Jalali (2019) [MM-1] and by Makkuva et al. (2019) [MM-2].

To assess the quality of the recovered transport map $\nabla\psi^\dagger$, we consider **unexplained variance percentage**: $\mathcal{L}^2\text{-UVP}(\nabla\psi^\dagger) = 100 \cdot \left[ \|\nabla\psi^\dagger - \nabla\psi^*\|_{\mathbb{P}}^2 / \text{Var}(\mathbb{Q}) \right]\%$. Here $\nabla\psi^*$ is the optimal transport map. For values $\approx 0\%$, $\nabla\psi^\dagger$ is a good approximation of OT map. For values $\geq 100\%$, map $\nabla\psi^\dagger$ is nearly useless. Indeed, a trivial benchmark $\nabla\psi^0(x) = \mathbb{E}_{\mathbb{Q}}[y]$ provides $\mathcal{L}^2\text{-UVP}(\nabla\psi^0) = 100\%$.

As it is seen from Table 1, LSOT leads to high error which grows drastically with the dimension. W2GN, MM-1 and MM-2 approaches perform nearly equally in metrics. It is expected since they both optimize analogous objectives. These methods compute optimal transport maps with low error ($\mathcal{L}^2\text{-UVP}<3\%$ even in $\mathbb{R}^{4096}$). However, as it is seen from the convergence plot in Figure 2, our approach **converges several times faster**: it naturally follows from the fact that MM approaches contain an inner optimization cycle. We discuss the experiment in more detail in Appendix C.4.

| Dim | 2 | 4 | 8 | 16 | 32 | 64 | 128 | 256 | 512 | 1024 | 2048 | 4096 |
|---|---|---|---|---|---|---|---|---|---|---|---|---|
| **LSOT** | <1 | 3.7 | 7.5 | 14.3 | 23 | 34.7 | 46.9 | | | >50 | | |
| **MM-1** | <1 | <1 | <1 | <1 | <1 | 1.2 | 1.4 | 1.3 | 1.5 | 1.6 | 1.8 | 2.7 |
| **MM-2** | <1 | <1 | <1 | <1 | <1 | <1 | 1 | 1.1 | 1.2 | 1.3 | 1.5 | 2.1 |
| **W2GN** | <1 | <1 | <1 | <1 | <1 | <1 | 1 | 1.1 | 1.3 | 1.3 | 1.8 | 1.5 |

Table 1: Comparison of $\mathcal{L}^2$-UVP (%) for LSOT, MM-1, MM-2 and W2GN (ours) methods in $D = 2, 4, \ldots, 2^{12}$.

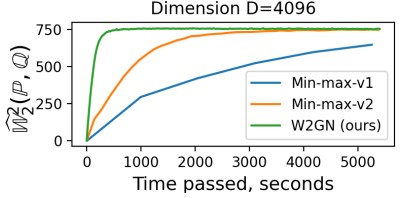

Figure 2: Comparison of convergence of W2GN (ours), MM-1, MM-2 approaches.

## 5.2 LATENT SPACE OPTIMAL TRANSPORT

We test our algorithm on **CelebA** image generation ($64 \times 64$). First, we construct the latent space distribution by using non-variational convolutional auto-encoder to encode the images to 128-dimensional latent vectors. Next, we use a pair of DenseIC-NNs to fit a cyclically monotone mapping to transform standard normal noise into the latent space distribution.

| Method | FID |
|---|---|
| AE: $Dec(Enc(X))$ | 7.5 |
| AE Raw Decode: $Dec(Z)$ | 31.81 |
| **W2GN+AE**: $Dec(g^\dagger(Z))$ | 17.21 |
| WGAN-QC : $Gen(Z)$ | 14.41 |

Table 2: FID scores for $64 \times 64$ generated images.

In Figure 3, we present images generated directly by sampling from standard normal noise before (1st row) and after (2nd row) applying out transport map. While our generative mapping does not perform significant changes, its effect is seen visually as well as confirmed by improvement of Frechet Inception Distance (FID, see Heusel et al. (2017)), see Table 2. For comparison, we also provide the score of a recent Wasserstein GAN by Liu et al. (2019). In Appendix C.5, we provide additional examples and the visualization of the latent space.

Decoded
$Z\sim N(0,I)$

Decoded
$g^\dagger(Z)$

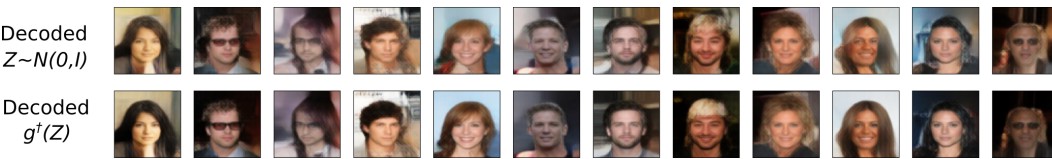

Figure 3: Images decoded from standard Gaussian latent noise (1st row) and decoded from the same noise transferred by our cycle monotone map (2nd row).

### 5.3 Image-to-Image Style Translation

In the problem of unpaired style transfer, the learner gets two image datasets, each with its own attributes, e.g. each dataset consists of landscapes related to a particular season. The goal is to fit a mapping capable of **transferring attributes** of one dataset to the other one, e.g. changing a winter landscape to a corresponding summer landscape.

The principal model for solving this problem is **CycleGAN** by Zhu et al. (2017). It uses 4 networks and optimizes a minimax objective to train the model. Our method uses **2 networks**, and has non-minimax objective.

We experiment with ConvICNN potentials on publicly availaible **Winter2Summer** and **Photo2Cezanne** datasets containing $256 \times 256$ pixel images. We train our model on mini-batches of randomly cropped $128 \times 128$ pixel RGB image parts. The results on Winter2Summer and Photo2Cezanne datasets applied to random $128 \times 128$ crops are shown in Figure 4.

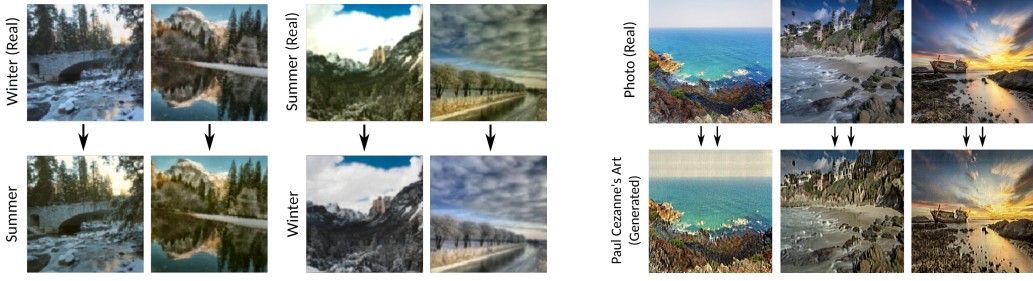

(a) Winter2Summer dataset results.                    (b) Photo2Cezanne dataset results.

Figure 4: Results of image-to-image style transfer by ConvICNN, $128 \times 128$ pixel images.

Our generative model fits a cycle monotone mapping. However, the desired style transfer may not be cycle monotone. Thus, our model may transfer only some of the required attributes. For example, for winter-to-summer transfer our model learned to colorize trees to green. Yet the model experiences problems with replacing snow masses with grass.

As Seguy et al. (2017) noted, OT is permutation invariant. It does not take into account the relation between dimensions, e.g. neighbouring pixels or channels of one pixel. Thus, OT may struggle to fit the optimal generative mapping via convolutional architectures (designed to preserve the local structure of the image). Figures 4a and 4b demonstrate highlights of our model. Yet we provide examples when the model does not perform well in Appendix C.8.

To fix the above mentioned issue, one may consider OT for the quadratic cost defined on the Gaussian pyramid of an image Burt & Adelson (1983) or, similar to perceptual losses used for super-resolution Johnson et al. (2016), consider perceptual quadratic cost. This statement serves as the challenge for our further research.

## 6 Conclusion

In this paper, we developed an end-to-end algorithm with a non-minimax objective for training cyclically monotone generative mappings, i.e. optimal transport mappings for the quadratic cost. Additionally, we established theoretical justification for our method from the approximation point of view. The results of computational experiments confirm the potential of the algorithm in various practical problems: latent space mass transport, image-to-image color/style transfer, domain adaptation.

### Acknowledgements

The work was partially supported by the *Russian Foundation for Basic Research grant 21-51-12005 NNIO_a*.

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

## A  PROOFS

In Subsection A.1, we provide important additional properties of Wasserstein-2 distance and related $\mathcal{L}^2$-spaces required to prove our main theoretical results.

In Subsection A.2, we prove our main Theorems 4.1 and 4.2. Additionally, we show how our proofs can be translated to the simpler case (Theorem A.3), i.e. optimizing correlations with a single potential (8) by using the basic approach of Taghvaei & Jalali (2019).

In Subsection A.3, we justify the pipeline of Figure 13. To do this, we prove a theorem that makes it possible to estimate the quality of the latent space generative mapping combined with the decoding part of the encoder.

In Subsection A.4, we prove a useful fact which makes it possible to apply our method to distributions which do not have a density.

### A.1  PROPERTIES OF WASSERSTEIN-2 METRIC AND RELATION TO $\mathcal{L}^2$ SPACES

To prove our results, we need to introduce **Kantorovich**'s formulation of Optimal Transport Kantorovitch (1958) which extends Monge's formulation (3). For a given transport cost $c : \mathcal{X} \times \mathcal{Y} \to \mathbb{R}$ and probability distributions $\mathbb{P}$ and $\mathbb{Q}$ on $\mathcal{X}$ and $\mathcal{Y}$ respectively, we define

$$\text{Cost}(\mathbb{P}, \mathbb{Q}) = \min_{\mu \in \Pi(\mathbb{P}, \mathbb{Q})} \int_{\mathcal{X} \times \mathcal{Y}} c(x, y) d\mu(x, y), \tag{17}$$

where $\Pi(\mathbb{P}, \mathbb{Q})$ is the set of all probability measures on $\mathcal{X} \times \mathcal{Y}$ whose marginals are $\mathbb{P}$ and $\mathbb{Q}$ respectively (**transport plans**). If the optimal transport solution exists in the form of mapping $g^* : \mathcal{X} \to \mathcal{Y}$ minimizing (3), then the optimal transport plan in (17) is given by $\mu^* = [\text{id}, g^*] \circ \mathbb{P}$. Otherwise, formulation (17) can be viewed as a relaxation of (3).

For quadratic cost $c(x, y) = \frac{1}{2}\|x - y\|^2$, the root of (17) defines Wasserstein-2 distance ($\mathbb{W}_2$), a **metric** on the space of probability distributions. In particular, it satisfies the **triangle inequality**, i.e. for every triplets of probability distributions $\mathbb{P}_1, \mathbb{P}_2, \mathbb{P}_3$ on $\mathcal{X} \subset \mathbb{R}^D$ we have

$$\mathbb{W}_2(\mathbb{P}_1, \mathbb{P}_3) \leq \mathbb{W}_2(\mathbb{P}_1, \mathbb{P}_2) + \mathbb{W}_2(\mathbb{P}_2, \mathbb{P}_3). \tag{18}$$

We will also need the following lemma.

**Lemma A.1** (Lipschitz property of Wasserstein-2 distance). *Let $\mathbb{P}, \mathbb{P}'$ be two probability distributions with finite second moments on $\mathcal{X}_1 \subset \mathbb{R}^{D_1}$. Let $T : \mathcal{X}_1 \to \mathcal{X}_2 \subset \mathbb{R}^{D_2}$ be a measurable mapping with Lipschitz constant bounded by $L$. Then the following inequality holds true:*

$$\mathbb{W}_2(T \circ \mathbb{P}, T \circ \mathbb{P}') \leq L \cdot \mathbb{W}_2(\mathbb{P}, \mathbb{P}'), \tag{19}$$

*i.e. the distribution distance between $\mathbb{P}, \mathbb{P}'$ mapped by $T$ does not exceed the initial distribution distance multiplied by Lipschitz constant $L$.*

*Proof.* Let $\mu^* \in \Pi(\mathbb{P}_1, \mathbb{P}_2)$ be the optimal transport plan between $\mathbb{P}_1$ and $\mathbb{P}_2$. Consider the distribution on $\mathcal{X}_2 \times \mathcal{X}_2$ given by $\mu = T \circ \mu^*$, where mapping $T$ is applied component-wise. The left and the right marginals of $\mu$ are equal to $T \circ \mathbb{P}_1$ and $T \circ \mathbb{P}_2$ respectively. Thus, $\mu \in \Pi(T \circ \mathbb{P}_1, T \circ \mathbb{P}_2)$ is a transport plan between $T \circ \mathbb{P}_1$ and $T \circ \mathbb{P}_2$. The cost of $\mu$ is not smaller than the optimal cost, i.e.

$$\mathbb{W}_2^2(T \circ \mathbb{P}_1, T \circ \mathbb{P}_2) \leq \int_{\mathcal{X}_2 \times \mathcal{X}_2} \frac{\|x - x'\|^2}{2} d\mu(x, x'). \tag{20}$$

Next, we use the Lipschitz property of $T$ and derive

$$\int_{\mathcal{X}_2 \times \mathcal{X}_2} \frac{\|x - x'\|^2}{2} d\mu(x, x') = \left[\mu = T \circ \mu^*\right] = \int_{\mathcal{X}_1 \times \mathcal{X}_1} \frac{\|T(x) - T(x')\|^2}{2} d\mu^*(x, x') \leq$$

$$\int_{\mathcal{X}_1 \times \mathcal{X}_1} \frac{L^2 \|x - x'\|^2}{2} d\mu^*(x, x') = L^2 \int_{\mathcal{X}_1 \times \mathcal{X}_1} \frac{\|x - x'\|^2}{2} d\mu^*(x, x') = L^2 \cdot \mathbb{W}_2^2(\mathbb{P}_1, \mathbb{P}_2). \quad (21)$$

To finish the proof, we combine (20) with (21) and obtain the desired inequality (19). $\qquad \square$

Throughout the rest of the paper, we use $\mathcal{L}^2(\mathcal{X} \to \mathbb{R}^D, \mathbb{P})$ to denote the **Hilbert space** of functions $f : \mathcal{X} \to \mathbb{R}^D$ with integrable square w.r.t. probability measure $\mathbb{P}$. The corresponding inner product for $f_1, f_2 \in \mathcal{L}^2(\mathcal{X} \to \mathbb{R}^D, \mathbb{P})$ is denoted by

$$\langle f_1, f_2 \rangle_{\mathbb{P}} \overset{def}{=} \int_{\mathcal{X}} \langle f_1(x), f_2(x) \rangle d\mathbb{P}(x).$$

We use $\|\cdot\|_{\mathbb{P}} = \sqrt{\langle \cdot, \cdot \rangle_{\mathbb{P}}}$ to denote the corresponding norm induced by the inner product.

**Lemma A.2** ($\mathcal{L}^2$ inequality for Wasserstein-2 distance). *Let $\mathbb{P}$ be a probability distribution on $\mathcal{X}_1 \subset \mathbb{R}^{D_1}$. Let $T_1, T_2 \in \mathcal{L}^2(\mathcal{X}_1 \to \mathbb{R}^{D_2}, \mathbb{P})$. Then the following inequality holds true:*

$$\frac{1}{2} \|T_1(x) - T_2(x)\|_{\mathbb{P}}^2 \geq \mathbb{W}_2^2(T_1 \circ \mathbb{P}, T_2 \circ \mathbb{P}).$$

*Proof.* We define the transport plan $\mu = [T_1, T_2] \circ \mathbb{P}$ between $T_1 \circ \mathbb{P}$ and $T_2 \circ \mathbb{P}$ and, similar to the previous Lemma A.1, use the fact that its cost is not smaller than the optimal cost. $\qquad \square$

### A.2 PROOFS OF THE MAIN THEORETICAL RESULTS

First, we prove our main Theorem 4.1. Then we formulate and prove its analogue (Theorem A.3) for the basic correlation optimization method (8) with single convex potential. Next, we prove our main Theorem 4.2. At the end of the subsection, we discuss the constants appearing in theorems: strong convexity and smoothness parameters.

*Proof of Theorem 4.1.* We split the proof into three subsequent parts.

**Part 1. Upper Bound on Correlations.**

First, we establish a lower bound for regularized correlations $\text{Corr}\big(\mathbb{P}, \mathbb{Q} \mid \psi^{\dagger}, \overline{\psi^{\ddagger}}; \lambda\big)$ omitting the regularization term.

$$\int_{\mathcal{X}} \psi^{\dagger}(x) d\mathbb{P}(x) + \int_{\mathcal{Y}} \big[ \langle y, \nabla \overline{\psi^{\ddagger}}(y) \rangle - \psi^{\dagger}\big(\nabla \overline{\psi^{\ddagger}}(y)\big) \big] d\mathbb{Q}(y) =$$

$$\int_{\mathcal{Y}} \psi^{\dagger}\big(\nabla \overline{\psi^*}(y)\big) d\mathbb{Q}(y) + \int_{\mathcal{Y}} \big[ \langle y, \nabla \overline{\psi^{\ddagger}}(y) \rangle - \psi^{\dagger}\big(\nabla \overline{\psi^{\ddagger}}(y)\big) \big] d\mathbb{Q}(y) = \quad (22)$$

$$\int_{\mathcal{Y}} \big[ \psi^{\dagger}\big(\nabla \overline{\psi^*}(y)\big) - \psi^{\dagger}\big(\nabla \overline{\psi^{\ddagger}}(y)\big) \big] d\mathbb{Q}(y) + \int_{\mathcal{Y}} \big[ \langle y, \nabla \overline{\psi^{\ddagger}}(y) \rangle \big] d\mathbb{Q}(y) \geq$$

$$\int_{\mathcal{Y}} \big[ \langle \nabla \psi^{\dagger} \circ \nabla \overline{\psi^{\ddagger}}(y), \nabla \overline{\psi^*}(y) - \nabla \overline{\psi^{\ddagger}}(y) \rangle + \frac{\beta^{\dagger}}{2} \|\nabla \overline{\psi^*}(y) - \nabla \overline{\psi^{\ddagger}}(y)\|^2 \big] d\mathbb{Q}(y) + \quad (23)$$

$$\int_{\mathcal{Y}} \langle y, \nabla \overline{\psi^{\ddagger}}(y) \rangle d\mathbb{Q}(y) + \Big[ \underbrace{\int_{\mathcal{Y}} \langle y, \nabla \overline{\psi^*}(y) \rangle d\mathbb{Q}(y)}_{\text{Corr}(\mathbb{P}, \mathbb{Q})} - \underbrace{\int_{\mathcal{Y}} \langle y, \nabla \overline{\psi^*}(x) \rangle d\mathbb{Q}(y)}_{\text{Corr}(\mathbb{P}, \mathbb{Q})} \Big] = \quad (24)$$

$$\langle \nabla \psi^{\dagger} \circ \nabla \overline{\psi^{\ddagger}}, \nabla \overline{\psi^*} - \nabla \overline{\psi^{\ddagger}} \rangle_{\mathbb{Q}} + \frac{\beta^{\dagger}}{2} \|\nabla \overline{\psi^*} - \nabla \overline{\psi^{\ddagger}}\|_{\mathbb{Q}}^2 - \langle \text{id}_{\mathcal{Y}}, \nabla \overline{\psi^*} - \nabla \overline{\psi^{\ddagger}} \rangle_{\mathbb{Q}} + \text{Corr}(\mathbb{P}, \mathbb{Q}) = \quad (25)$$

$$\langle \nabla \psi^{\dagger} \circ \nabla \overline{\psi^{\ddagger}} - \text{id}_{\mathcal{Y}}, \nabla \overline{\psi^*} - \nabla \overline{\psi^{\ddagger}} \rangle_{\mathbb{Q}} + \frac{\beta^{\dagger}}{2} \|\nabla \overline{\psi^*} - \nabla \overline{\psi^{\ddagger}}\|_{\mathbb{Q}}^2 + \text{Corr}(\mathbb{P}, \mathbb{Q}) =$$

$$\frac{1}{2\beta^{\dagger}} \|\nabla \psi^{\dagger} \circ \nabla \overline{\psi^{\ddagger}} - \text{id}_{\mathcal{Y}}\|_{\mathbb{Q}} + \langle \nabla \psi^{\dagger} \circ \nabla \overline{\psi^{\ddagger}} - \text{id}_{\mathcal{Y}}, \nabla \overline{\psi^*} - \nabla \overline{\psi^{\ddagger}} \rangle_{\mathbb{Q}} + \frac{\beta^{\dagger}}{2} \|\nabla \overline{\psi^*} - \nabla \overline{\psi^{\ddagger}}\|_{\mathbb{Q}}^2 +$$

$$\text{Corr}(\mathbb{Q}, \mathbb{P}) - \frac{1}{2\beta^{\dagger}} \|\nabla \psi^{\dagger} \circ \nabla \overline{\psi^{\ddagger}} - \text{id}_{\mathcal{Y}}\|_{\mathbb{Q}}^2 =$$

$$\text{Corr}(\mathbb{Q}, \mathbb{P}) +$$

$$\frac{1}{2}\left\|\frac{1}{\sqrt{\beta^\dagger}}\big[\nabla\psi^\dagger \circ \nabla\overline{\psi^\ddagger} - \text{id}_{\mathcal{Y}}\big] + \sqrt{\beta^\dagger}\big[\nabla\overline{\psi^*} - \nabla\overline{\psi^\ddagger}\big]\right\|_{\mathbb{Q}}^2 - \frac{1}{2\beta^\dagger}\|\nabla\psi^\dagger \circ \nabla\overline{\psi^\ddagger} - \text{id}_{\mathcal{Y}}\|_{\mathbb{Q}}^2. \quad (26)$$

In transition to line (22), we use change of variables formula $\mathbb{P} = \nabla\overline{\psi^*} \circ \mathbb{Q}$. In line (23), we use $\beta^\dagger$-strong convexity of function $\psi^\dagger$ and then add zero term in line (24). Next, for simplicity, we replace integral notation with $\mathcal{L}^2(\mathcal{Y} \to \mathbb{R}^D, \mathbb{Q})$ notation starting from line (25).

We add the omitted regularization term back to (26) and obtain the following bound:

$$\text{Corr}\big(\mathbb{P}, \mathbb{Q} \mid \psi^\dagger, \overline{\psi^\ddagger}; \lambda\big) \geq \text{Corr}(\mathbb{Q}, \mathbb{P}) + \frac{1}{2}(\lambda - \frac{1}{\beta^\dagger}) \cdot \|\nabla\psi^\dagger \circ \nabla\overline{\psi^\ddagger} - \text{id}_{\mathcal{Y}}\|_{\mathbb{Q}}^2 +$$

$$\frac{1}{2}\left\|\frac{1}{\sqrt{\beta^\dagger}}\big[\nabla\psi^\dagger \circ \nabla\overline{\psi^\ddagger} - \text{id}_{\mathcal{Y}}\big] + \sqrt{\beta^\dagger}\big[\nabla\overline{\psi^*} - \nabla\overline{\psi^\ddagger}\big]\right\|_{\mathbb{Q}}^2. \quad (27)$$

Since $\lambda > \frac{1}{\beta^\dagger}$, the obtained inequality proves that the true correlations $\text{Corr}(\mathbb{P}, \mathbb{Q})$ are upper bounded by the regularized correlations $\text{Corr}\big(\mathbb{P}, \mathbb{Q} \mid \psi^\dagger, \overline{\psi^\ddagger}; \lambda\big)$. Note that if the optimal map $\nabla\psi^\dagger$ is $\geq \beta^\dagger$ strongly convex, the bound (27) is tight. Indeed, it turns into equality when we substitute $\nabla\overline{\psi^\ddagger} = (\nabla\psi^\dagger)^{-1} = \nabla\overline{\psi^*}$.

### Part 2. Inverse Generative Property.

We continue the derivations of part 1. Let $u = \nabla\psi^\dagger \circ \nabla\overline{\psi^\ddagger} - \text{id}_{\mathcal{Y}}$ and $v = \nabla\overline{\psi^*} - \nabla\overline{\psi^\ddagger}$. By matching (27) with (14), we obtain

$$\epsilon \geq \frac{1}{2}(\lambda - \frac{1}{\beta^\dagger})\|u\|_{\mathbb{Q}}^2 + \frac{1}{2}\left\|\frac{1}{\sqrt{\beta^\dagger}}u + \sqrt{\beta^\dagger}v\right\|_{\mathbb{Q}}^2. \quad (28)$$

Now we derive an upper bound for $\|v\|_{\mathbb{Q}}^2$. For a fixed $u$ we have

$$\left\|\frac{1}{\sqrt{\beta^\dagger}}u + \sqrt{\beta^\dagger}v\right\|_{\mathbb{Q}}^2 \leq 2\epsilon - (\lambda - \frac{1}{\beta^\dagger})\|u\|_{\mathbb{Q}}^2.$$

Next, we apply the triangle inequality:

$$\|\sqrt{\beta^\dagger}v\|_{\mathbb{Q}} \leq \left\|\frac{1}{\sqrt{\beta^\dagger}}u + \sqrt{\beta^\dagger}v\right\|_{\mathbb{Q}} + \|\frac{1}{\sqrt{\beta^\dagger}}u\|_{\mathbb{Q}} \leq \sqrt{2\epsilon - (\lambda - \frac{1}{\beta^\dagger})\|u\|_{\mathbb{Q}}^2} + \|\frac{1}{\sqrt{\beta^\dagger}}u\|_{\mathbb{Q}}. \quad (29)$$

The expression of the right-hand side of (29) attains its maximal value $\sqrt{\frac{2\epsilon}{1 - \frac{1}{\lambda\beta^\dagger}}}$ at $\|u\|_{\mathbb{Q}} = \sqrt{\frac{2\epsilon}{\lambda^2\beta^\dagger - \lambda}}$. We conclude that

$$\|\nabla\overline{\psi^*} - \nabla\overline{\psi^\ddagger}\|_{\mathbb{Q}}^2 = \|v\|_{\mathbb{Q}}^2 \leq \frac{2\epsilon}{\beta^\dagger - \frac{1}{\lambda}}.$$

Finally, we apply $\mathcal{L}^2$-inequality of Lemma A.2 to distribution $\mathbb{Q}$, mappings $\nabla\overline{\psi^*}$ and $\nabla\overline{\psi^\ddagger}$, and obtain $\mathbb{W}_2^2(\nabla\overline{\psi^\ddagger} \circ \mathbb{P}, \mathbb{Q}) \leq \frac{\epsilon}{\beta^\dagger - \frac{1}{\lambda}}$, i.e. the desired upper bound on the distance between the generated and target distribution.

### Part 3. Forward Generative Property.

We recall the bound (28). Since all the summands are positive, we derive

$$\|u\|_{\mathbb{Q}}^2 \leq \frac{2\epsilon}{\lambda - \frac{1}{\beta^\dagger}}. \quad (30)$$

We will use (30) to obtain an upper bound on $\|\nabla\psi^* - \nabla\psi^\dagger\|_{\mathbb{P}}$.

To begin with, we note that since $\psi^\dagger$ is $\beta^\dagger$-strongly convex, its convex conjugate $\overline{\psi^\dagger}$ is $\frac{1}{\beta^\dagger}$-smooth. Thus, gradient $\nabla\overline{\psi^\dagger}$ is $\frac{1}{\beta^\dagger}$-Lipschitz. We conclude that for all $x, x' \in \mathcal{X}$:

$$\|\nabla\overline{\psi^\dagger}(x) - \nabla\overline{\psi^\dagger}(x')\| \leq \frac{1}{\beta^\dagger}\|x - x'\|. \tag{31}$$

We raise both parts of (31) into the square, substitute $x = \nabla\psi^\dagger \circ \nabla\overline{\psi^\dagger}(y)$ and $x' = \nabla\psi^\dagger \circ \nabla\overline{\psi^\ddagger}(y)$ and integrate over $\mathcal{Y}$ w.r.t $\mathbb{Q}$. The obtained inequality is as follows:

$$\int_{\mathcal{Y}} \|\nabla\overline{\psi^\dagger}(y) - \nabla\overline{\psi^\ddagger}(y)\|^2 d\mathbb{Q}(y) \leq \frac{1}{(\beta^\dagger)^2}\int_{\mathcal{Y}}\|\nabla\psi^\dagger \circ \nabla\overline{\psi^\dagger}(y) - \nabla\psi^\dagger \circ \nabla\overline{\psi^\ddagger}(y)\|^2 d\mathbb{Q}(y) \tag{32}$$

Next, we derive

$$\|\nabla\overline{\psi^\dagger} - \nabla\overline{\psi^\ddagger}\|^2_{\mathbb{Q}} = \int_{\mathcal{Y}}\|\nabla\overline{\psi^\dagger}(y) - \nabla\overline{\psi^\ddagger}(y)\|^2 d\mathbb{Q}(y) \leq$$

$$\int_{\mathcal{Y}}\frac{1}{(\beta^\dagger)^2}\|\overbrace{\nabla\psi^\dagger \circ \nabla\overline{\psi^\dagger}(y)}^{=y} \underbrace{- \nabla\psi^\dagger \circ \nabla\overline{\psi^\ddagger}(y)}_{=-u(y)}\|^2 d\mathbb{Q}(y) = \frac{\|u\|^2_{\mathbb{Q}}}{(\beta^\dagger)^2}. \tag{33}$$

In transition to line (33), we use the previously obtained inequality (32).

Next, we use the triangle inequality for $\|\cdot\|_{\mathbb{Q}}$ to bound

$$\|\nabla\overline{\psi^\dagger} - \nabla\overline{\psi^*}\|_{\mathbb{Q}} \leq \|\nabla\overline{\psi^\dagger} - \nabla\overline{\psi^\ddagger}\|_{\mathbb{Q}} + \|\underbrace{\nabla\overline{\psi^\ddagger} - \nabla\overline{\psi^*}}_{=v}\|_{\mathbb{Q}} \leq$$

$$\sqrt{\frac{2\epsilon}{\lambda - \frac{1}{\beta^\dagger}}}\cdot\frac{1}{\beta^\dagger} + \sqrt{\frac{2\epsilon}{\beta^\dagger - \frac{1}{\lambda}}} = \sqrt{\frac{2\epsilon}{\lambda - \frac{1}{\beta^\dagger}}}\cdot(\frac{1}{\beta^\dagger} + \sqrt{\frac{\lambda}{\beta^\dagger}}) = \sqrt{\frac{2\epsilon}{\lambda\beta^\dagger - 1}}\cdot(\frac{1}{\sqrt{\beta^\dagger}} + \sqrt{\lambda}) \tag{34}$$

Next, we derive a lower bound for the left-hand side of (34) by using $\mathcal{B}^\dagger$-smoothness of $\psi^\dagger$. For all $x, x' \in \mathcal{X}$ we have

$$\|\nabla\psi^\dagger(x) - \nabla\psi^\dagger(x')\| \leq \mathcal{B}^\dagger\|x - x'\|. \tag{35}$$

We raise both parts of (35) to the square, substitute $x = \nabla\overline{\psi^\dagger}(y)$ and $x' = \nabla\overline{\psi^*}(y)$, and integrate over $\mathcal{Y}$ w.r.t. $\mathbb{Q}$:

$$\int_{\mathcal{Y}}\|\nabla\psi^\dagger \circ \nabla\overline{\psi^\dagger}(y) - \nabla\psi^\dagger \circ \nabla\overline{\psi^*}(y)\|^2 d\mathbb{Q}(y) \leq (\mathcal{B}^\dagger)^2\int_{\mathcal{Y}}\|\nabla\overline{\psi^\dagger}(y) - \nabla\overline{\psi^*}(y)\|^2 d\mathbb{Q}(y) \tag{36}$$

Next, we use (36) to derive

$$\|\nabla\overline{\psi^\dagger} - \nabla\overline{\psi^*}\|^2_{\mathbb{Q}} = \int_{\mathcal{Y}}\|\nabla\overline{\psi^\dagger}(y) - \nabla\overline{\psi^*}(y)\|^2 d\mathbb{Q}(y) \geq$$

$$\int_{\mathcal{Y}}\frac{1}{(\mathcal{B}^\dagger)^2}\|\underbrace{\nabla\psi^\dagger \circ \nabla\overline{\psi^\dagger}(y)}_{=y} - \nabla\psi^\dagger \circ \nabla\overline{\psi^*}(y)\|^2 d\mathbb{Q}(y) =$$

$$\frac{1}{(\mathcal{B}^\dagger)^2}\int_{\mathcal{Y}}\|\nabla\psi^*(x) - \nabla\psi^\dagger(x)\|^2 d\mathbb{P}(x) \geq \frac{2}{(\mathcal{B}^\dagger)^2}\mathbb{W}^2_2(\nabla\psi^\dagger \circ \mathbb{P}, \mathbb{Q}) \tag{37}$$

In line (37), we use the $\mathcal{L}^2$ property of Wasserstein-2 distance (Lemma A.2). We conclude that

$$\mathbb{W}^2_2(\nabla\psi^\dagger \circ \mathbb{P}, \mathbb{Q}) \leq \frac{(\mathcal{B}^\dagger)^2 \cdot \epsilon}{\lambda\beta^\dagger - 1}\cdot\Big(\frac{1}{\sqrt{\beta^\dagger}} + \sqrt{\lambda}\Big)^2,$$

and finish the proof. □

It is quite straightforward to formulate analogous result for the basic optimization method with single potential (8). We summarise the statement in the following

**Theorem A.3** (Generative Property for Approximators of Correlations). *Let* $\mathbb{P}, \mathbb{Q}$ *be two continuous probability distributions on* $\mathcal{Y} = \mathcal{X} = \mathbb{R}^D$ *with finite second moments. Let* $\psi^* : \mathcal{Y} \to \mathbb{R}$ *be the convex minimizer of* $\mathrm{Corr}(\mathbb{P}, \mathbb{Q}|\psi)$.

*Let differentiable* $\psi^\dagger$ *is* $\beta^\dagger$-*strongly convex and* $\mathcal{B}^\dagger$-*smooth* $(\mathcal{B}^\dagger \geq \beta^\dagger > 0)$ *function* $\psi^\dagger : \mathcal{X} \to \mathbb{R}$ *satisfy*

$$\mathrm{Corr}\big(\mathbb{P}, \mathbb{Q} \mid \psi^\dagger\big) \leq \left[ \int_{\mathcal{X}} \psi^*(x)d\mathbb{P}(x) + \int_{\mathcal{Y}} \overline{\psi^*}(y)d\mathbb{Q}(y) \right] + \epsilon = \mathrm{Corr}(\mathbb{P}, \mathbb{Q}) + \epsilon. \qquad (38)$$

*Then the following inequalities hold true:*

1. **Forward Generative Property** *(map* $g^\dagger = \nabla\psi^\dagger$ *pushes* $\mathbb{P}$ *to be* $O(\epsilon)$-*close to* $\mathbb{Q}$*)*
$$\mathbb{W}_2^2(g^\dagger \circ \mathbb{P}, \mathbb{Q}) = \mathbb{W}_2^2(\nabla\psi^\dagger \circ \mathbb{P}, \mathbb{Q}) \leq \mathcal{B}^\dagger\epsilon;$$

2. **Inverse Generative Property** *(map* $(g^\dagger)^{-1} = \nabla\overline{\psi^\dagger} = (\nabla\psi^\dagger)^{-1}$ *pushes* $\mathbb{Q}$ *to be* $O(\epsilon)$-*close to* $\mathbb{P}$*)*
$$\mathbb{W}_2^2\big((g^\dagger)^{-1} \circ \mathbb{Q}, \mathbb{P}\big) \leq \frac{\epsilon}{\beta^\dagger}.$$

*Proof.* First, we note that $\epsilon \geq 0$ by the definition of $\psi^*$. Next, we repeat the first part of the proof of Theorem (4.1) by substituting $\overline{\psi^\ddagger} := \overline{\psi^\dagger}$. Thus, by using $\nabla\psi^\dagger \circ \nabla\overline{\psi^\ddagger} = \mathrm{id}_{\mathcal{Y}}$ we obtain the following simple analogue of formula (27):

$$\mathrm{Corr}(\mathbb{P}, \mathbb{Q} \mid \psi^\dagger) \geq \mathrm{Corr}(\mathbb{P}, \mathbb{Q}) + \frac{\beta^\dagger}{2}\|\nabla\overline{\psi^*} - \nabla\overline{\psi^\dagger}\|_{\mathbb{Q}}, \qquad (39)$$

i.e. $\epsilon \geq \frac{\beta^\dagger}{2}\|\nabla\overline{\psi^*} - \nabla\overline{\psi^\dagger}\|_{\mathbb{Q}}$. Thus, by using Lemma (A.2), we immediately derive $\mathbb{W}_2^2(\nabla\overline{\psi^\dagger} \circ \mathbb{Q}, \mathbb{P}) \leq \frac{\epsilon}{\beta^\dagger}$, i.e. inverse generative property.

To derive forward generative property, we note that $\mathcal{B}^\dagger$-smoothness of $\psi^\dagger$ means that $\overline{\psi^\dagger}$ is $\frac{1}{\mathcal{B}^\dagger}$-strongly convex. Thus, due to symmetry of the objective, we can repeat all the derivations w.r.t. $\overline{\psi^\dagger}$ instead of $\psi^\dagger$ in order to prove
$$\mathbb{W}_2^2(\nabla\psi^\dagger \circ \mathbb{P}, \mathbb{Q}) \leq \mathcal{B}^\dagger\epsilon,$$
i.e. forward generative property. $\qquad\qquad\square$

*Proof of Theorem 4.2.* We repeat the first part of the proof of Theorem 4.1, but instead of exploiting strong convexity of $\psi^{\mathcal{X}}$ to obtain an upper bound on the regularized correlations, we use $\mathcal{B}^{\mathcal{X}}$-smoothness to obtain a lower bound. The resulting analogue (40) to (27) is as follows:

$$\mathrm{Corr}\big(\mathbb{P}, \mathbb{Q} \mid \psi^{\mathcal{X}}, \overline{\psi^{\mathcal{Y}}}, \lambda\big) - \mathrm{Corr}(\mathbb{P}, \mathbb{Q}) \leq$$
$$\frac{1}{2}(\lambda - \frac{1}{\mathcal{B}^{\mathcal{X}}}) \cdot \|\nabla\psi^{\mathcal{X}} \circ \nabla\overline{\psi^{\mathcal{Y}}} - \mathrm{id}_{\mathcal{Y}}\|_{\mathbb{Q}}^2 +$$
$$\frac{1}{2}\left\| \frac{1}{\sqrt{\mathcal{B}^{\mathcal{X}}}}\big[\nabla\psi^{\mathcal{X}} \circ \nabla\overline{\psi^{\mathcal{Y}}} - \mathrm{id}_{\mathcal{Y}}\big] + \sqrt{\mathcal{B}^{\mathcal{X}}}\big[\nabla\overline{\psi^*} - \nabla\overline{\psi^{\mathcal{Y}}}\big]\right\|_{\mathbb{Q}}^2 \leq \quad(40)$$
$$\frac{1}{2}(\lambda - \frac{1}{\mathcal{B}^{\mathcal{X}}}) \cdot \|\nabla\psi^{\mathcal{X}} \circ \nabla\overline{\psi^{\mathcal{Y}}} - \mathrm{id}_{\mathcal{Y}}\|_{\mathbb{Q}}^2 +$$
$$\frac{1}{2}\left[ \frac{1}{\sqrt{\mathcal{B}^{\mathcal{X}}}}\|\nabla\psi^{\mathcal{X}} \circ \nabla\overline{\psi^{\mathcal{Y}}} - \mathrm{id}_{\mathcal{Y}}\|_{\mathbb{Q}} + \sqrt{\mathcal{B}^{\mathcal{X}}}\|\nabla\overline{\psi^*} - \nabla\overline{\psi^{\mathcal{Y}}}\|_{\mathbb{Q}} \right]^2 = \quad(41)$$
$$\frac{\lambda}{2} \cdot \|\nabla\psi^{\mathcal{X}} \circ \nabla\overline{\psi^{\mathcal{Y}}} - \mathrm{id}_{\mathcal{Y}}\|_{\mathbb{Q}}^2 + \|\nabla\psi^{\mathcal{X}} \circ \nabla\overline{\psi^{\mathcal{Y}}} - \mathrm{id}_{\mathcal{Y}}\|_{\mathbb{Q}} \cdot \|\nabla\overline{\psi^*} - \nabla\overline{\psi^{\mathcal{Y}}}\|_{\mathbb{Q}} + \frac{\mathcal{B}^{\mathcal{X}}}{2}\|\nabla\overline{\psi^*} - \nabla\overline{\psi^{\mathcal{Y}}}\|_{\mathbb{Q}}^2 \,(42)$$

Here in transition from line (40) to (41), we apply the triangle inequality.

For every $y \in \mathcal{Y}$ we have $\|\nabla\psi^{\mathcal{X}} \circ \nabla\overline{\psi^{\mathcal{Y}}}(y) - \nabla\psi^{\mathcal{X}} \circ \nabla\overline{\psi^*}(y)\| \leq \mathcal{B}^{\mathcal{X}} \cdot \|\nabla\overline{\psi^{\mathcal{Y}}}(y) - \nabla\overline{\psi^*}(y)\|$. We raise both parts of this inequality to the power 2 and integrate over $\mathcal{Y}$ w.r.t. $\mathbb{Q}$. We obtain

$$\|\nabla\psi^{\mathcal{X}} \circ \nabla\overline{\psi^{\mathcal{Y}}} - \nabla\psi^{\mathcal{X}} \circ \nabla\overline{\psi^*}\|_{\mathbb{Q}}^2 \leq (\mathcal{B}^{\mathcal{X}})^2 \cdot \|\nabla\overline{\psi^{\mathcal{Y}}} - \nabla\overline{\psi^*}\|_{\mathbb{Q}}^2 \leq (\mathcal{B}^{\mathcal{X}})^2 \cdot \epsilon_{\mathcal{Y}}. \qquad (43)$$

Now we recall that since $\overline{\nabla\psi^*} \circ \mathbb{Q} = \mathbb{P}$, we have

$$\|\nabla\psi^{\mathcal{X}} \circ \nabla\overline{\psi^*} - \underbrace{\nabla\psi^* \circ \nabla\overline{\psi^*}}_{\mathrm{id}_{\mathcal{Y}}}\|_{\mathbb{Q}}^2 = \|\nabla\psi^{\mathcal{X}} - \nabla\psi^*\|_{\mathbb{P}}^2 \leq \epsilon_{\mathcal{X}}. \tag{44}$$

Next, we combine (43) and (44) apply the triangle inequality to bound

$$\|\nabla\psi^{\mathcal{X}} \circ \nabla\overline{\psi^{\mathcal{Y}}} - \mathrm{id}_{\mathcal{Y}}\|_{\mathbb{Q}} \leq$$
$$\|\nabla\psi^{\mathcal{X}} \circ \nabla\overline{\psi^{\mathcal{Y}}}(y) - \nabla\psi^{\mathcal{X}} \circ \nabla\overline{\psi^*}(y)\|_{\mathbb{Q}} + \|\nabla\psi^{\mathcal{X}} \circ \nabla\overline{\psi^*} - \underbrace{\nabla\psi^* \circ \nabla\overline{\psi^*}}_{\mathrm{id}_{\mathcal{Y}}}\|_{\mathbb{Q}} \leq$$
$$\mathcal{B}^{\mathcal{X}}\sqrt{\epsilon_{\mathcal{Y}}} + \sqrt{\epsilon_{\mathcal{X}}}. \tag{45}$$

We substitute all the bounds to (40):

$$\mathrm{Corr}\big(\mathbb{P}, \mathbb{Q} \mid \psi^{\mathcal{X}}, \overline{\psi^{\mathcal{Y}}}, \lambda\big) - \mathrm{Corr}(\mathbb{P}, \mathbb{Q}) \leq$$
$$\frac{\lambda}{2}(\mathcal{B}^{\mathcal{X}}\sqrt{\epsilon_{\mathcal{Y}}} + \sqrt{\epsilon_{\mathcal{X}}})^2 + (\mathcal{B}^{\mathcal{X}}\sqrt{\epsilon_{\mathcal{Y}}} + \sqrt{\epsilon_{\mathcal{X}}}) \cdot (\sqrt{\epsilon_{\mathcal{Y}}}) + \frac{\mathcal{B}^{\mathcal{X}}}{2}\epsilon_{\mathcal{Y}}. \tag{46}$$

and finish the proof by using

$$\mathrm{Corr}\big(\mathbb{P}, \mathbb{Q} \mid \psi^{\dagger}, \overline{\psi^{\ddagger}}, \lambda\big) \leq \mathrm{Corr}\big(\mathbb{P}, \mathbb{Q} \mid \psi^{\mathcal{X}}, \overline{\psi^{\mathcal{Y}}}, \lambda\big)$$

which follows from the definition of $\psi^{\dagger}, \overline{\psi^{\ddagger}}$. $\qquad\square$

One may formulate and prove analogous result for the basic optimization method with a single potential (8). However, we do not include this in the paper since a similar result exists Taghvaei & Jalali (2019).

All our theoretical results require **smoothness** or **strong convexity** properties of potentials. We note that the assumption of smoothness and strong convexity also appears in other papers on Wasserstein-2 optimal transport, see e.g. Paty et al. (2019).

The property of $\mathcal{B}$-smoothness of a convex function $\psi$ means that its gradient $\nabla\psi$ has Lipshitz constant bounded by $\mathcal{B}$. In our case, constant $\mathcal{B}$ serves as a reasonable measure of complexity of the fitted mapping $\nabla\psi$: it estimates how much the mapping can warp the space.

Strong convexity is dual to smooothness in the sense that a convex conjugate $\overline{\psi}$ to $\beta$-strongly convex function $\psi$ is $\frac{1}{\beta}$-smooth (and vise-versa) Kakade et al. (2009). In our case, $\beta$-strongly convex potential means that its inverse gradient mapping $(\nabla\psi)^{-1} = \nabla\overline{\psi}$ can not significantly warp the space, i.e. has Lipshitz constant bounded by $\frac{1}{\beta}$.

Recall the setting of our Theorem 4.2. Assume that the optimal transport map $\nabla\psi^*$ between $\mathbb{P}$ and $\mathbb{Q}$ is a gradient of $\beta$-strongly convex ($\beta > 0$) and $\mathcal{B}$-smooth ($\mathcal{B} < \infty$) function. In this case, by considering classes $\Psi_{\mathcal{X}} = \overline{\Psi}_{\mathcal{Y}}$ equal to all $\min(\beta, \frac{1}{\mathcal{B}})$-strongly convex and $\max(\mathcal{B}, \frac{1}{\beta})$-smooth functions, by using our method (for any $\lambda > \frac{1}{\beta}$) we will exactly compute correlations and find the optimal $\nabla\psi^*$.

## A.3 FROM LATENT SPACE TO DATA SPACE

In the setting of the latent space mass transport, we fit a generative mapping to the latent space of an autoencoder and combine it with the decoder to obtain a generative model. The natural question is how close decoded distribution is to the real data distribution $\mathbb{S}$ used to train an encoder. The following Theorem states that the distribution distance of the combined model can be naturally divided into two parts: the quality of the latent fit and the reconstruction loss of the auto-encoder.

**Theorem A.4** (Decoding Theorem). *Let $\mathbb{S}$ be the real data distribution on $\mathcal{S} \subset \mathbb{R}^K$. Let $u : \mathcal{S} \to \mathcal{Y} = \mathbb{R}^D$ be the encoder and $v : \mathcal{Y} \to \mathbb{R}^K$ be $L$-Lipschitz decoder.*

*Assume that a latent space generative model has fitted a map $g^{\dagger} : \mathcal{X} \to \mathcal{Y}$ that pushes some latent distribution $\mathbb{P}$ on $\mathcal{X} = \mathbb{R}^D$ to be $\epsilon$ close to $\mathbb{Q} = u \circ \mathbb{S}$ in $\mathbb{W}_2^2$-sense, i.e.*

$$\mathbb{W}_2^2(g^{\dagger} \circ \mathbb{P}, \mathbb{Q}) \leq \epsilon.$$

*Then the following inequality holds true:*

$$\mathbb{W}_2\big(\underbrace{v \circ g^\dagger \circ \mathbb{P}}_{\substack{\text{Generated data}\\\text{distribuion}}}, \mathbb{S}\big) \leq L\sqrt{\epsilon} + \big(\frac{1}{2}\underbrace{\mathbb{E}_{\mathbb{S}}\|s - v \circ u(s)\|_2^2}_{\substack{\text{Autoencoder's}\\\text{reconstruction loss}}}\big)^{\frac{1}{2}}, \qquad (47)$$

*where $v \circ g^\dagger$ is the combined generative model.*

*Proof.* We apply the triangle inequality and obtain

$$\mathbb{W}_2(v \circ g^\dagger \circ \mathbb{P}, \mathbb{S}) \leq \mathbb{W}_2(v \circ g^\dagger \circ \mathbb{P}, v \circ \mathbb{Q}) + \mathbb{W}_2(v \circ \mathbb{Q}, \mathbb{S}). \qquad (48)$$

Let $\mathbb{P}^\dagger = g^\dagger \circ \mathbb{P}$ be the fitted latent distribution ($\mathbb{W}_2^2(\mathbb{P}^\dagger, \mathbb{Q}) \leq \epsilon$). We use Lipschitz Wasserstein-2 property of Lemma A.1 and obtain

$$L\sqrt{\epsilon} \geq L \cdot \mathbb{W}_2(\mathbb{P}^\dagger, \mathbb{Q}) \geq \mathbb{W}_2(v \circ \mathbb{P}^\dagger, v \circ \mathbb{Q}) = \mathbb{W}_2(v \circ g^\dagger \circ \mathbb{P}, v \circ \mathbb{Q}). \qquad (49)$$

Next, we apply $\mathcal{L}^2$-property (Lemma A.2) to mappings $\text{id}_{\mathbb{S}}$, $v \circ u$ and distribution $\mathbb{S}$, and derive

$$\frac{1}{2}\mathbb{E}_{\mathbb{S}}\|s - v \circ u(s)\|_2^2 \geq \mathbb{W}_2^2(\mathbb{S}, v \circ \mathbb{Q}). \qquad (50)$$

The desired inequality (47) immediately results from combining (48) with (49), (50). $\qquad\square$

## A.4    EXTENSION TO THE NON-EXISTENT DENSITY CASE

Our main theoretical results require distributions $\mathbb{P}, \mathbb{Q}$ to have finite second moments and density on $\mathcal{X} = \mathcal{Y} = \mathbb{R}^D$. While the existence of second moments is a reasonable condition, in the majority of practical use-cases the density might not exist. Moreover, it is typically assumed that the supports of distributions are manifold of dimension lower than $D$ or even discrete sets.

One may artificially smooth distributions $\mathbb{P}, \mathbb{Q}$ by convolving them with a random white Gaussian noise[6] $\Lambda = \mathcal{N}(0, \sigma^2 I_D)$ and find a generative mapping $g^\dagger : \mathcal{X} \to \mathcal{Y}$ between smoothed $\mathbb{P} * \Lambda$ and $\mathbb{Q} * \Lambda$. For Wasserstein-2 distances, it is natural that the generative properties of $g^\dagger$ as a mapping between $\mathbb{P} * \Lambda$ and $\mathbb{Q} * \Lambda$ will transfer to generative properties of $g^\dagger$ as a mapping between $\mathbb{P}, \mathbb{Q}$, but with some bias depending on the statistics of $\Lambda$.

**Theorem A.5** (De-smoothing Wasserstein-2 Property). *Let $\mathbb{P}, \mathbb{Q}$ be two probability distributions on $\mathcal{X} = \mathcal{Y} = \mathbb{R}^D$ with finite second moments. Let $\Lambda = \mathcal{N}(0, \sigma^2 I_D)$ be a Gaussian white noise . Let $\mathbb{P} * \Lambda$ and $\mathbb{Q} * \Lambda$ be versions of $\mathbb{P}$ and $\mathbb{Q}$ smoothed by $\Lambda$. Let $T : \mathcal{X} \to \mathcal{Y}$ be a L-Lipschitz measurable map satisfying*

$$\mathbb{W}_2(T \circ [\mathbb{P} * \Lambda], [\mathbb{Q} * \Lambda]) \leq \sqrt{\epsilon}.$$

*Then the following inequality holds true:*

$$\mathbb{W}_2(T \circ \mathbb{P}, \mathbb{Q}) \leq (L+1)\sigma\sqrt{\frac{D}{2}} + \sqrt{\epsilon}.$$

*Proof.* We apply the triangle inequality twice and obtain

$$\mathbb{W}_2(T \circ \mathbb{P}, \mathbb{Q}) \leq \mathbb{W}_2(T \circ \mathbb{P}, T \circ [\mathbb{P} * \Lambda]) + \mathbb{W}_2(T \circ [\mathbb{P} * \Lambda], [\mathbb{Q} * \Lambda]) + \mathbb{W}_2([\mathbb{Q} * \Lambda], \mathbb{Q}).$$

Consider a transport plan $\mu(y, y') \in \Pi([\mathbb{Q} * \Lambda], \mathbb{Q})$ satisfying $\mu(y' \mid y) = \Lambda(y' - y)$. The cost of $\mu$ is given by

$$\int_{\mathcal{Y}} \int_{\mathcal{Y}} \frac{\|y - y'\|^2}{2} d\Lambda(y - y') d\mathbb{Q}(y) = \int_{\mathcal{Y}} \frac{D\sigma^2}{2} d\mathbb{Q}(y) = \frac{D\sigma^2}{2}.$$

Since the plan is not necessarily optimal, we conclude that $\mathbb{W}_2^2([\mathbb{Q} * \Lambda], \mathbb{Q}) \leq \frac{D\sigma^2}{2}$. Analogously, we conclude that $\mathbb{W}_2^2(\mathbb{P}, [\mathbb{P} * \Lambda]) \leq \frac{D\sigma^2}{2}$. Next, we apply Lipschitz property of Wasserstein-2 (Lemma A.1) and obtain:

$$\mathbb{W}_2(T \circ \mathbb{P}, T \circ [\mathbb{P} * \Lambda]) \leq L \cdot \mathbb{W}_2(\mathbb{P}, [\mathbb{P} * \Lambda]) = L\sigma\sqrt{\frac{D}{2}}.$$

---

[6]From the practical point of view, smoothing is equal to adding random noise distributed according to $\Lambda$ to samples from $\mathbb{P}, \mathbb{Q}$ respectively.

Finally, we combine all the obtained bounds and derive

$$\mathbb{W}_2(T \circ \mathbb{P}, \mathbb{Q}) \le (L+1)\sigma\sqrt{\frac{D}{2}} + \sqrt{\epsilon}.$$

□

## B   NEURAL NETWORK ARCHITECTURES

In Subsection B.1, we describe the general architecture of the input convex networks. In this section, we describe particular realisations of the general architecture that we use in experiments: **DenseICNN** in Subsection B.2 and **ConvICNN** in Subsection B.3.

### B.1   GENERAL INPUT-CONVEX ARCHITECTURE

We approximate convex potentials by Input Convex Neural Networks Amos et al. (2017). The overall architecture is schematically presented in Figure 5.

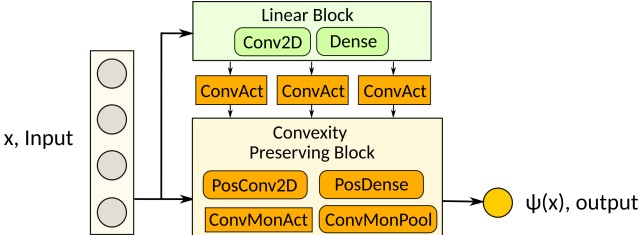

Figure 5: General architecture of an Input Convex Neural Network.

Input convex network consists of **two** principal **blocks**:

1. **Linear (L) block** consists of linear layers. Activation functions and pooling operators in the block are also linear, e.g. identity activation or average pooling.
2. **Convexity preserving (CP) block** consists of linear layers with **non-negative weights** (excluding biases). Activations and pooling operators in this block are **convex** and **monotone**.

Within blocks it is possible to use arbitrary **skip connections** obeying the stated rules. Neurons of L Block can be arbitrarily connected to those of CP block by applying a convex activation[7] and adding the result with a positive weight. It comes from the convex function arithmetic that every neuron (including the output one) in the architecture of Figure 5 is a convex function of the input.[8]

In our case, we expect the network to be able to easily fit the **identity generative mapping**

$$g(x) = \nabla\psi(x) = x,$$

i.e. $\psi(x) = \frac{1}{2}\|x\|^2 + c$ is a quadratic function. Thus, we mainly insert **quadratic activations** between L and CP blocks, which differs from Amos et al. (2017) where no activation was used. Gradients of input quadratic functions correspond to linear warps of the input and are intuitively highly useful as building blocks (in particular, for fitting identity mapping).

We use specific architectures which fit to the general scheme shown in Figure 5. **ConvICNN** is used for image-processing tasks, and **DenseICNN** is used otherwise. The exact architectures are described in the subsequent subsections.

We use **CELU** function as a convex and monotone activation (within CP block) in all the networks. We have also tried **SoftPlus** among some other continuous and differentiable functions, yet this

---

[7]Unlike activations within convexity preserving block, convex activation between L and CP block may not be monotone, e.g. $\sigma(x) = x^2$ can be used as an activation.

[8]It is possible into insert **batch norm** and **dropout** to L and CP blocks as well as between them. These layers do not affect convexity since they can be considered (during inference) as linear layers with non-negative weights.

negatively impacted the performance. The usage of **ReLU** function is also possible, but the gradient of the potential in this case will be discontinuous. Thus, it will not be Lipschitz, and the insights of our Theorems 4.1 and 4.2 may not work.

As a convex and monotone pooling (within CP block), it is possible to use **Average** and **LogSumExp** pooling (smoothed max pooling). Pure **Max** pooling should be avoided for the same reason as **ReLU** activation. However, in ConvICNN architecture we use convolutions with stride instead of pooling, see Subsection B.3.

In order to use insights of Theorem 4.1, we impose strong convexity and smoothness on the potentials. As we noted in Appendix A.2, $\mathcal{B}$-**smoothness** of a convex function is equal to $\frac{1}{\mathcal{B}}$ **strong convexity** of its conjugate function (and vise versa). Thus, we make both networks $\psi_\theta, \overline{\psi_\omega}$ to be $\beta := \frac{1}{\mathcal{B}}$ strongly convex, and cycle regularization keeps $\lesssim \frac{1}{\beta} = \mathcal{B}$ smoothness for $\psi_\theta \approx (\overline{\psi_\omega})^{-1}$ and $\overline{\psi_\omega} \approx (\overline{\psi_\theta})^{-1}$. In practice, we achieve strong convexity by adding extra value $\frac{\beta}{2}\|x\|^2$ to the output of the final neuron of a network. In all our experiments, we set $\beta^{-1} = 1000000$.[9] In addition to smoothing, strong convexity guarantees that $\nabla\psi_\theta$ and $\nabla\overline{\psi_\omega}$ are bijections, which is used in Theorems 4.1, 4.2.

### B.2 DENSE INPUT CONVEX NEURAL NETWORK

For DenseICNN, we implement **Convex Quadratic** layer each output neuron of which is a convex quadratic function of input. More precisely, for each input $x \in \mathbb{R}^{N_{\text{in}}}$ it outputs $(\text{cq}_1(x), \ldots, \text{cq}_{N_{\text{out}}}(x)) \in \mathbb{R}^{N_{\text{out}}}$, with

$$\text{cq}_n(x) = \langle x, A_n x \rangle + \langle b_n, x \rangle + c_n$$

for positive semi-definite quadratic form $A \in \mathbb{R}^{N_{\text{in}} \times N_{\text{in}}}$, vector $b \in \mathbb{R}^{N_{\text{in}}}$ and constant $c \in \mathbb{R}$. Note that for large $N_{\text{in}}$, the size of such layer grows fast, i.e. $\geq O(N_{\text{in}}^2 \cdot N_{\text{out}})$. To fix this issue, we represent each quadratic matrix as a product $A_n = F_n^T F_n$, where $F \in \mathbb{R}^{r \times N_{\text{in}}}$ is the matrix of rank at most $r$. This helps to limit optimization to only positive quadratic forms (and, in particular, symmetric), and reduce the number of weights stored for quadratic part to $O(r \cdot N_{\text{in}} \cdot N_{\text{out}})$. Actually, the resulting quadratic forms $A_n$ will have rank at most $r$.

The architecture is shown in Figure 6. We use Convex Quadratic Layers in DenseICNN for connecting input directly to layers of a fully connected network. Note that such layers (even at full rank) do not blow the size of the network when the input dimension is low, e.g. in the problem of color transfer.

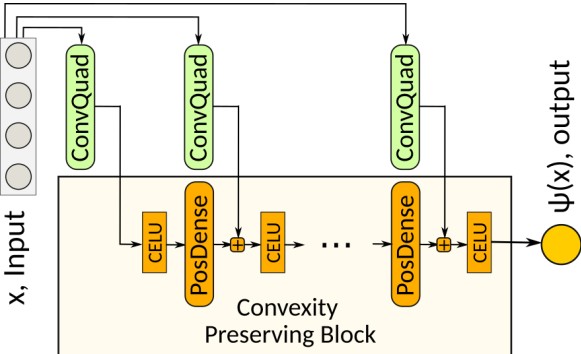

Figure 6: Dense Input Convex Neural Network.

The hyperparameters of DenseICNN are widths of the layers and ranks of the convex input-quadratic layers. For simplicity, we use the same rank $r$ for all the layers. We denote the width of the first convex quadratic layer by $h_0$ and the width of $k+1$-th Convex Quadratic and $k$-th Linear layers by $h_k$. The complete hyperparameter set of the network is given by $[r; h_0; h_1, \ldots, h_K]$.

---

[9]Imposing smoothness & strong convexity can be viewed as a regularization of the mapping: it does not perform too large/small warps of the input. See e.g. Paty et al. (2019).

### B.3 Convolutional Input Convex Neural Network

We apply convolutional networks to the problem of unpaired image-to-image style transfer. The architecture of ConvICNN is shown in Figure 7. The network takes an input image ($128 \times 128$ with 3 RGB channels) and outputs a single value. The gradient of the network w.r.t. the input serves as a generator in our algorithm.

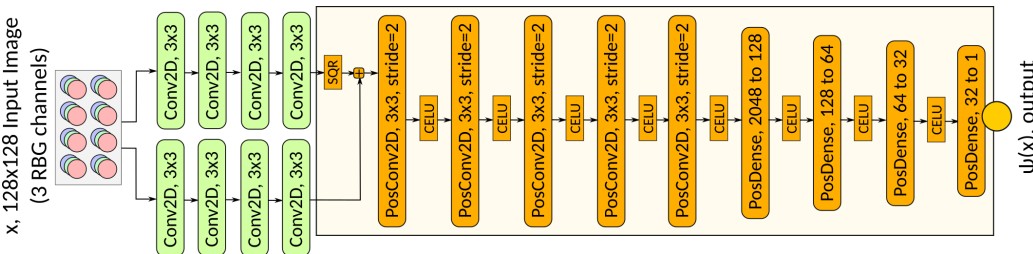

Figure 7: Convolutional Input Convex Neural Network. All convolutional layers have 128 channels.

Linear and Convexity preserving blocks are successive, and no skip connections are used. Block L consists of two separate parts with stacked convolutions without intermediate activation. The square of the second part is added to the first part and is used as an input for the CP block. All convolutional layers of the network have 128 channels (zero-padding with offset $= 1$ is used).

## C Experimental Details and Extra Results

In the first subsection, we describe general training details. In the second subsection, we discuss the computational complexity of a single gradient step of our method. Each subsequent subsection corresponds to a particular problem and provides additional experimental results and training details: toy experiments in Subsection C.3 and comparison with minimax approach in Subsection C.4, latent space optimal transport in Subsection C.5, image-to-image color transfer in C.6, domain adaptation in Subsection C.7, image-to-image style transfer in Subsection C.8.

### C.1 General Training Details

The code is written on **PyTorch** framework. The networks are trained on a single **GTX 1080Ti**. The numerical optimization procedure is provided in Algorithm 1 of Section 4.1.

In each experiment, both primal $\psi_\theta$ and conjugate $\overline{\psi_\omega}$ potentials have the same network architecture. The minimization of (12) is done via mini batch stochastic gradient descent with **weight clipping** (excluding biases) in CP block to the $[0, +\infty)$.[10] We use **Adam** Kingma & Ba (2014) optimizer.

For every particular task we pretrain the potential network $\psi_\theta$ by minimizing mean squared error to satisfy $\nabla \psi_\theta(x) \approx x$ and copy the weights to $\overline{\psi_\omega}$. This provides a good initialization for the main training, i.e. $\nabla \psi_\theta$ and $\nabla \overline{\psi_\omega}$ are mutually inverse.

Doing tests, we noted that our method converges faster if we disable back-propagation through term $\nabla \overline{\psi_\omega}$ which appears twice in the second line of (12). In this case, the derivative w.r.t. $\omega$ is computed by using the regularization terms only.[11] This heuristic allows to save additional memory and computational time because a smaller computational graph is built. We used the heuristic in all the experiments.

In the experiments with high dimensional data (latent space optimal transport, domain adaptation and style transfer), we add the following extra regularization term to the main objective (12):

$$R_\mathcal{X}(\theta, \omega) = \int_\mathcal{X} \|g_\omega^{-1} \circ g_\theta(x) - x\|^2 d\mathbb{P}(x) = \int_\mathcal{Y} \|\nabla \overline{\psi_\omega} \circ \nabla \psi_\theta(x) - x\|^2 d\mathbb{P}(x). \tag{51}$$

---

[10] We also tried to use softplus, exponent on weights and regularization instead of clipping, but none of these worked well.

[11] The term $\langle \nabla \overline{\psi_\omega}(y), y \rangle$ becomes redundant for the optimization. Yet it remains useful for monitoring the convergence.

Term (51) is analogous to the term $R_{\mathcal{Y}}(\theta, \omega)$ given by (11). It also keeps forward $g_\theta$ and inverse $g_\omega^{-1}$ generative mappings being approximately inverse. From the theoretical point of view, it is straightforward to obtain approximation guarantees similar to those of Theorems 4.1 and 4.2 for the optimization with two terms: $R_{\mathcal{X}}$ and $R_{\mathcal{Y}}$. However, we do not to include $R_{\mathcal{X}}$ in the proofs in order to keep them simple.

## C.2  COMPUTATIONAL COMPLEXITY

The time required to evaluate the value of (12) and its gradient w.r.t. $\theta, \omega$ is comparable up to a constant factor to that of a single evaluation of $\psi_\theta(x)$.

This claim follows from the well-known fact that gradient evaluation $\nabla_\theta h_\theta(x)$ of $h_\theta : \mathbb{R}^D \to \mathbb{R}$, when parameterized as a neural network, requires time proportional to the size of the computational graph. Hence gradient computation requires computational time proportional to the time for evaluating the function $h_\theta(x)$ itself. The same holds true when computing the derivative with respect to $x$. Thus, the number of operations required to compute different terms in (12), e.g. $\nabla\overline{\psi_\omega}(y)$, $\psi_\theta\big(\nabla\overline{\psi_\omega}(y)\big)$ and $\nabla\psi_\theta \circ \nabla\overline{\psi_\omega}(y)$, is also linear w.r.t. the computation time of $\psi_\theta(x)$ or, equivalently, $\overline{\psi_\omega}(x)$. As a consequence, the time required for the forward pass of (12) is larger than the forward pass for $\psi_\theta(x)$ only up to a constant factor. Thus, the backward pass for (12) with respect to parameters of ICNNs $\theta$ and $\omega$ is also linear in the computation time of $\psi_\theta(x)$.

We empirically measured that for our DenseICNN potentials, the computation of gradient of (12) w.r.t. parameters $\theta, \omega$ requires roughly 8-12x more time than the computation of $\psi_\theta(x)$. Evaluating $\nabla\psi_\theta, \nabla\overline{\psi_\omega}$ takes roughly 3-4x more time than evaluating $\psi_\theta(x)$.

## C.3  TOY EXPERIMENTS

In this subsection, we test our algorithm on $2D$ toy distributions from Gulrajani et al. (2017); Seguy et al. (2017). In all the experiments, distribution $\mathbb{P}$ is the standard Gaussian noise and $\mathbb{Q}$ is a Gaussian mixture or a Swiss roll.

Both primal and conjugate potentials $\psi_\theta$ and $\overline{\psi_\omega}$ have DenseICNN [2; 128; 128, 64] architecture. Each network has roughly 25000 trainable parameters. Some of them vanish during the training because of the weight clipping. For each particular problem the networks are trained for 30000 iterations with 1024 samples in a mini batch. Adam optimizer Kingma & Ba (2014) with lr $= 10^{-3}$ is used. We put $\lambda = 1$ in our cycle regularization and impose additional $10^{-10}$ $\mathcal{L}^1$ regularization on the weights.

For the case when $\mathbb{Q}$ is a mixture of 8 Gaussians, the intermediate learned distributions are shown in Figure 8. The overall structure of the forward mapping has already been learned on iteration 200, while the inverse mapping gets learned only on iteration $\approx 2000$. This can be explained by the smoothness of the desired optimal mappings $\nabla\psi^*$ and $\nabla\overline{\psi^*}$. The inverse mapping $\nabla\overline{\psi^*}$ has large Lipschitz constant because it has to unsqueeze dense masses of 8 Gaussians. In contrast to the inverse mapping, the forward mapping $\nabla\psi^*$ has to squeeze the distribution. Thus, it is expected to have lower Lipschitz constant (everywhere except the neighbourhood of a central point which is a fixed point of $\nabla\psi^*$ due to symmetry).

Additional examples (Gaussian Mixtures & Swiss roll) are shown in Figures 10a, 10b and 9. When $\mathbb{Q}$ is a mixture of 100 gaussians (Figure 9), our model learns all of the modes and does not suffer from **mode dropping**. We do not state that the fit is perfect but emphasize that **mode collapse** also does not happen.

Note that all our theoretical results require distributions $\mathbb{P}, \mathbb{Q}$ to be smooth. Our method fits **continuous** optimal transport map via **differentiable** w.r.t. input ICNNs with CELU activation. At the same time, Makkuva et al. (2019) also uses ICNNs with **discontinuous gradient** w.r.t. the input, e.g. by using ReLU activations to fit discontinuous generative mappings. We do not know whether our theoretical results can be directly generalized to the discontinuous mappings case (without using smoothing as we suggested in Subsection A.4). However, we note that the usage of ICNNs with discontinuous gradient naturally leads to "torn" generated distributions, see an example in Figure 11. While the fitted mapping is indeed close to the true Swiss Roll in $\mathbb{W}_2$ sense, it clearly suffers from

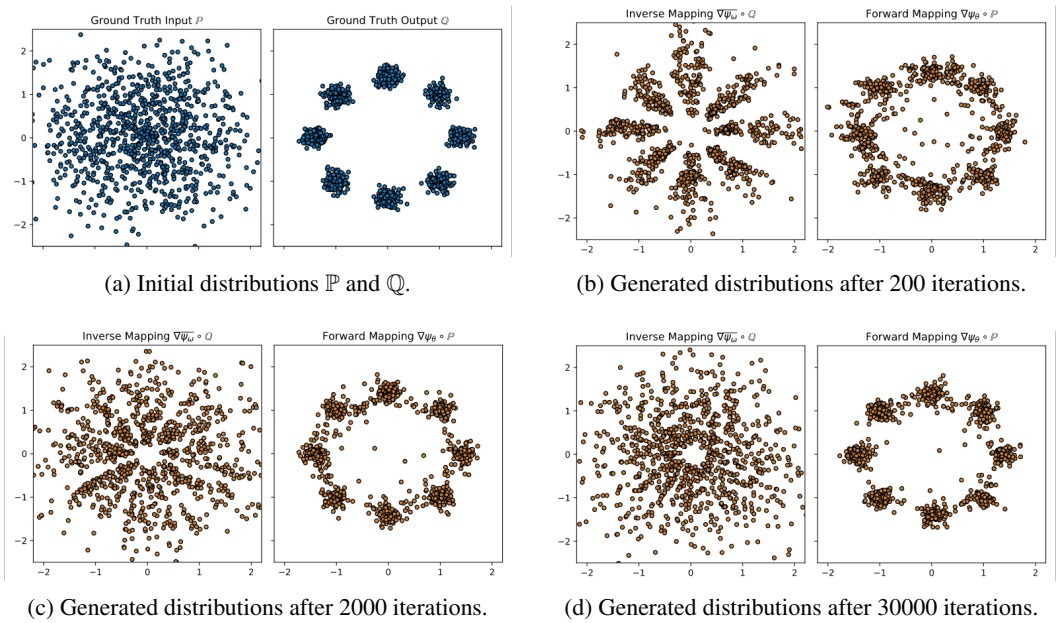

(a) Initial distributions $\mathbb{P}$ and $\mathbb{Q}$.

(b) Generated distributions after 200 iterations.

(c) Generated distributions after 2000 iterations.

(d) Generated distributions after 30000 iterations.

Figure 8: Convergence stages of our algorithm applied to fitting cycle monotone mappings (forward and inverse) between distributions $\mathbb{P}$ (Gaussian) and $\mathbb{Q}$ (Mixture of 8 Gaussians).

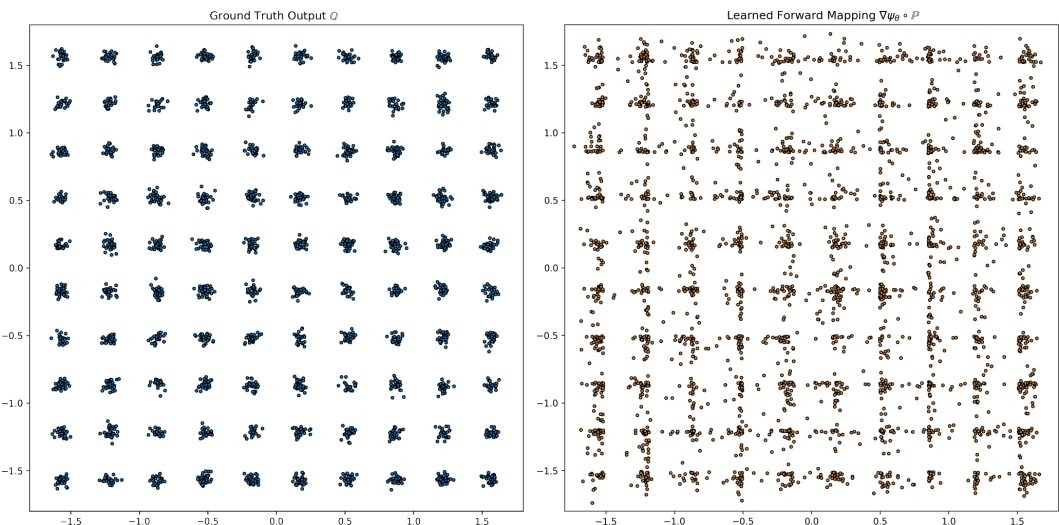

Figure 9: Mixture of 100 Gaussians $\mathbb{Q}$ and distribution $\nabla \psi_\theta \circ \mathbb{P}$ fitted by our algorithm.

"torn" effect. From the practical point of view, this effect seems to be similar to **mode collapse**, a well-known disease of GANs.

## C.4 GAUSSIAN OPTIMAL TRANSPORT DETAILS AND DISCUSSION

We consider the Gaussian setting $\mathbb{P}, \mathbb{Q} = \mathcal{N}(0, \Sigma_\mathbb{P}), \mathcal{N}(0, \Sigma_\mathbb{Q})$ for which the ground truth OT solution has a closed form, see (Álvarez-Esteban et al., 2016, Theorem 2.3). Considering non-centered $\mathbb{P}, \mathbb{Q}$ is unnecessary since $\mathbb{W}_2^2(\mathbb{P}, \mathbb{Q}) = \|\mu_\mathbb{P} - \mu_\mathbb{Q}\|^2 + \mathbb{W}_2^2(\mathbb{P}_0, \mathbb{Q}_0)$, where $\mathbb{P}_0, \mathbb{Q}_0$ are centered copies of $\mathbb{P}, \mathbb{Q}$. In $D$-dimensional space, $\sqrt{\Sigma_\mathbb{P}}$ ($\sqrt{\Sigma_\mathbb{Q}}$ - analogously) is initialized as $S_\mathbb{P}^T \Lambda S_\mathbb{P}$, where $S_\mathbb{P} \in O_D$ is a random rotation, $\Lambda$ is diagonal with eigenvalues $[\frac{1}{2}, \ldots, \frac{1}{2} b^k, \ldots, 2]$, $b = \sqrt[D-1]{4}$. The ground

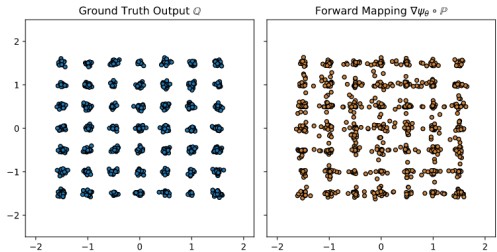
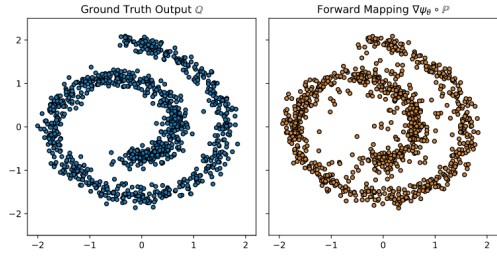

(a) Mixture of 49 Gaussians $\mathbb{Q}$ and distribution $\nabla\psi_\theta \circ \mathbb{P} \approx \mathbb{Q}$ fitted by our algorithm.

(b) Swiss Roll distribution $\mathbb{Q}$ and distribution $\nabla\psi_\theta \circ \mathbb{P} \approx \mathbb{Q}$ fitted by our algorithm.

Figure 10: Toy distributions fitted by our algorithm.

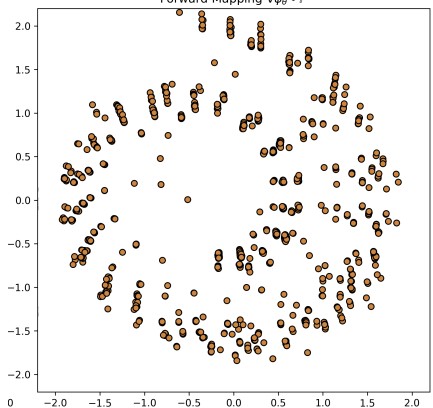

Figure 11: An example of a "torn" generative mapping to a Swiss Roll by a gradient of ICNN with ReLU activations.

truth optimal transport map from $\mathbb{P}$ to $\mathbb{Q}$ is linear and given by

$$\nabla\psi^*(x) = \Sigma_{\mathbb{P}}^{-\frac{1}{2}}\left(\Sigma_{\mathbb{P}}^{\frac{1}{2}}\Sigma_{\mathbb{Q}}\Sigma_{\mathbb{P}}^{\frac{1}{2}}\right)^{\frac{1}{2}}\Sigma_{\mathbb{P}}^{-\frac{1}{2}}x.$$

We compare our approach with the method by Seguy et al. (2017) [LSOT] and the minimax approaches by Taghvaei & Jalali (2019) [MM-1], Makkuva et al. (2019) [MM-2]. We recall the details of LSOT's regularized optimization of OT maps and distances. The objective is given by

$$\min_{\psi_\theta,\overline{\psi_\omega}}\left[\int_{\mathcal{X}}\psi_\theta(x)d\mathbb{P}(x)+\int_{\mathcal{Y}}\overline{\psi_\omega}(y)d\mathbb{Q}(y)+\frac{1}{2\epsilon}\int_{\mathcal{X}\times\mathcal{Y}}\left[\langle x,y\rangle-\psi_\theta(x)-\overline{\psi_\omega}(y)\right]_+^2 d\big(\mathbb{P}\times\mathbb{Q}\big)(x,y)\right],$$

where we defined $\psi_\theta = \frac{\|x\|^2}{2} - u_\theta(x)$ and $\overline{\psi_\omega} = \frac{\|y\|^2}{2} - v_\omega(y)$ to make LSOT notation $(u_\theta, v_\omega)$ match our notation $(\psi_\theta, \overline{\psi_\omega})$. We use L2 regularizer (but not entropy) since it empirically works better (as noted in LSOT paper). Potentials $\psi_\theta, \overline{\psi_\omega}$ are NOT restricted to be convex. The transport plan $\hat{T} \circ \mathbb{P} \approx \mathbb{Q}$ is recovered via the barycentric projection, see (Seguy et al., 2017, Section 4).

We set $\lambda = \min(D, 50)$ for our method and $\epsilon = 0.01$ for LSOT (chosen empirically). In all the methods we use DenseICNN$[1; D, D, \frac{D}{2}]$ of Subsection B.2. In LSOT, we do not convexify nets (do not clamp weights), i.e. they are "usual" unrestricted neural networks (empirically selected as the best option) as originally implied in LSOT paper.

It follows from the Table 1 in Section 5.1 that LSOT leads to high bias error which grows drastically with the dimension. While theoretically $\epsilon \to 0$ should solve the bias problem, practically small $\epsilon$ leads to optimization instabilities. LSOT's drawback is that estimation of the regularizer requires sampling from joint measure $\mathbb{P} \times \mathbb{Q}$ on $\mathbb{R}^{2D}$. For the majority of pairs $(x, y)$ the L2 regularizer vanishes (the effect worsens with $D \to \infty$). In contrast to LSOT, our cycle regularizer uses samples only from marginal measures ($\mathbb{Q}$ or $\mathbb{P}$) on $\mathbb{R}^D$. The biasing effect is studied theoretically (our Theorems 4.1, 4.2), and the bias is **inexistent** when optimal potentials $\psi^*, \overline{\psi^*}$ are contained in the approximating function classes.

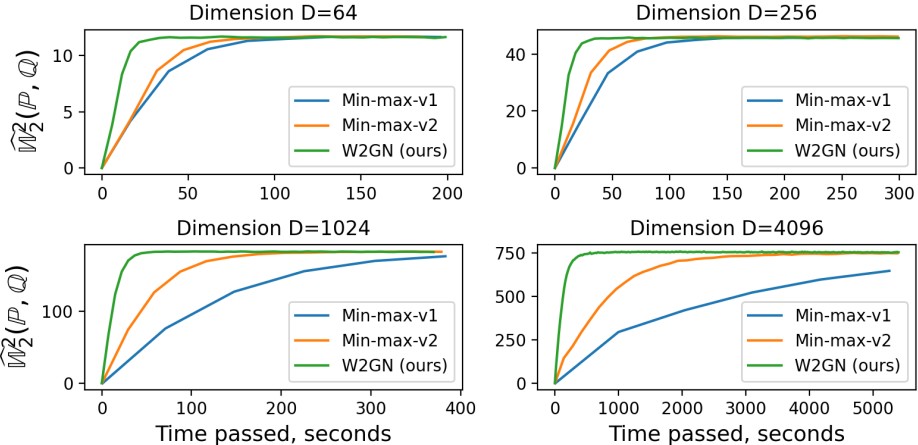

Figure 12: Comparison of convergence speed of W2GN, MM-1 and MM-2 approaches in dimensions $D = 64, 256, 1024, 4096$.

W2GN (ours), MM-1 and MM-2 methods are capable of computing maps and distances with low error ($\mathcal{L}^2$-UVP<3% even in $\mathbb{R}^{4096}$). However, as it is seen from the convergence plots in Figure 12, our approach **converges several times faster**: it naturally follows from the fact that MM-1, MM-2 approaches contain an inner optimization cycle.

## C.5 Latent Space Optimal Transport Details

We follow the pipeline of Figure 13 below. The latent space distribution is constructed by using convolution auto-encoder to encode **CelebA** images into 128-dimensional latent vectors respectively. To train the auto-encoder, we use a perceptual loss on features of a pre-trained VGG-16 network.

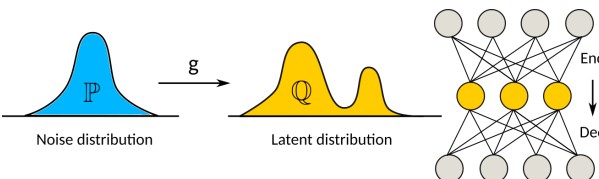

Figure 13: The pipeline of latent space mass transport.

We use DenseICNN [4; 256; 256; 128; 64] to fit a cyclically monotone generative mapping to transform standard normal noise into the latent space distribution. For each problem the networks are trained for 100000 iterations with 128 samples in a mini batch. Adam optimizer with lr $= 3 \times 10^{-4}$ is used. We put $\lambda = 100$ as the cycle regularization parameter.

We provide additional examples of generated images in Figure 14. We also visualize the latent space distribution of the autoencoder and the distribution fitted by generative map in Figure 15. The FID scores presented in Table 2 are computed via PyTorch implementation of FID Score[12]. As a benchmark score we added WGAN-QC by Liu et al. (2019).

Finally, we emphasize that cyclically monotone generative mapping that we fit is **explicit**. Similarly to Normalizing Flows Rezende & Mohamed (2015) and in contrast to other methods, such as Lei et al. (2019), it provides **tractable density** inside the latent space. Since $\nabla \psi_\omega \approx (\nabla \psi_\theta)^{-1}$ is differentiable and **injective**, one may use the change of variables formula for density $q(y) = [\det \nabla^2 \psi_\omega(y)] \cdot p(\nabla \psi_\omega(y))$ to study the latent space distribution.

---

[12]https://github.com/mseitzer/pytorch-fid

Decoded
*Z~N(0,I)*

Decoded
*g†(Z)*

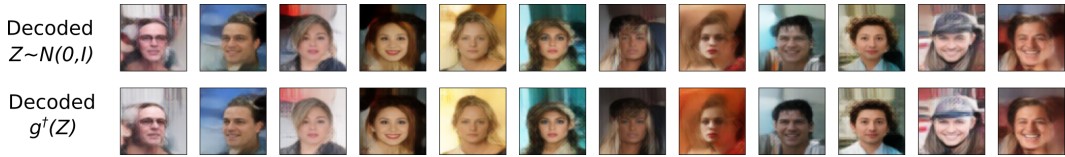

Figure 14: Images decoded from standard Gaussian latent noise (1st row) and decoded from the same noise transferred by our cycle monotone map (2nd row).

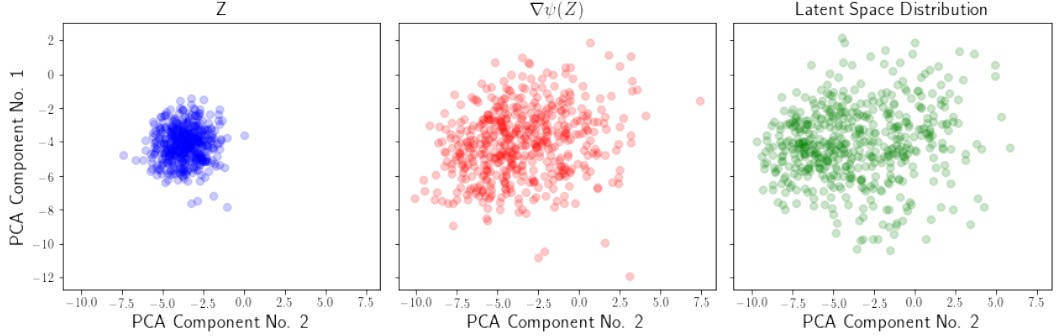

Figure 15: A pair of main principal components of CelebA Autoencoder's latent space. From left to right: $Z \sim \mathcal{N}(0, I)$ [blue], mapped $Z$ by W2GN [red], true autoencoders latent space [green]. PCA decomposition is fitted on autoencoders latent space [green].

## C.6 COLOR TRANSFER

The problem of color transfer between images[13] is to map the **color palette** of the image into the other one in order to make it look and "feel" similar to the original.

Optimal transport can be applied to color transfer, but it is sensitive to noise and outliers. To avoid these problems, several relaxations were proposed Rabin et al. (2014); Paty et al. (2019). These approaches solve a discrete version of Wasserstein-2 OT problem. The computation of optimal transport cost for large images is barely feasible or infeasible at all due to extreme size of color palettes. Thus, the **reduction of pixel color palette** by $k$-means clustering is usually performed to make OT computation feasible. Yet such a reduction may lose color information.

Our algorithm uses mini-batch stochastic optimization. Thus, it has no limitations on the size of color palettes. On training, we sequentially input mini-batches of images' pixels ($\in \mathbb{R}^3$) into potential networks with DenceICNN [3; 128; 128, 64] architecture.[14] The networks are trained for 5000 iterations with 1024 pixels in a mini batch. Adam optimizer with lr $= 10^{-3}$ is used. We put $\lambda = 3$ as the cycle regularization parameter. We impose extra $10^{-10}$ $\mathcal{L}^1$-penalty on the weights.

The color transfer results for $\approx 10$ megapixel images are presented in Figure 16a. The corresponding color palettes are given in Figure 16b. Additional example of color transfer are given in Figure 17.

## C.7 DOMAIN ADAPTATION

The domain adaptation problem is to learn a model $f$ (e.g. a classifier) from a source distribution $\mathbb{Q}$. This model has to perform well on a different (related) target distribution $\mathbb{P}$.

Most of the methods based on OT theory solve domain adaptation explicitly by transforming distribution $\mathbb{P}$ into $\mathbb{Q}$ and then applying the model $f$ to generated samples. In some cases the mapping $g : \mathcal{X} \to \mathcal{Y}$ (which transforms $\mathbb{P}$ to $\mathbb{Q}$) is obtained by solving a discrete OT problem Courty et al.

---

[13]Images may have unequal size. Yet they are assumed to have the same number of channels, e.g. RGB ones.

[14]Since our model is **parametric**, the complexity of fitted generative mapping $g^\dagger : \mathbb{R}^3 \to \mathbb{R}^3$ between the palettes depends on the size of potential networks.

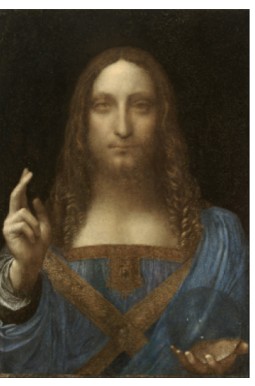 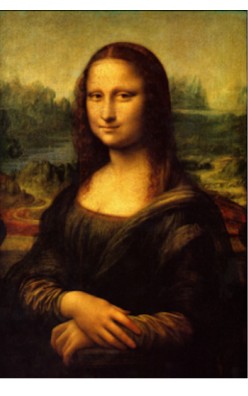 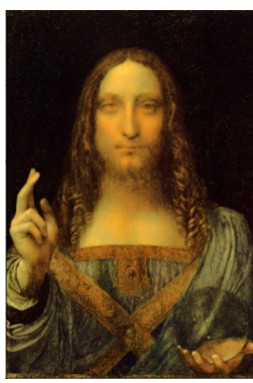 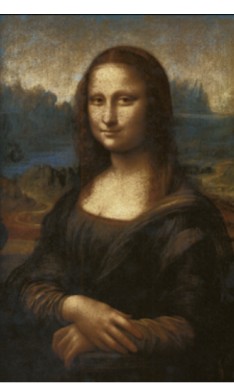

(a) Original images (on the left) and images obtained by color transfer (on the right). The sizes of images are $3300 \times 4856$ (first) and $2835 \times 4289$ (second).

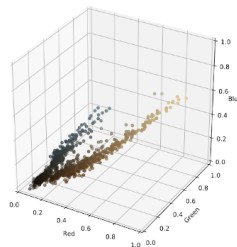 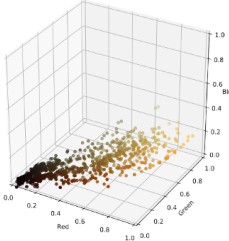 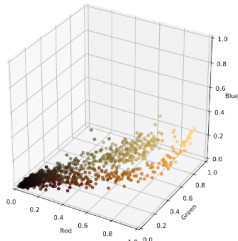 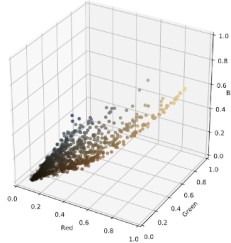

(b) Color palettes (3000 random pixels, best viewed in color) for the original images (on the left) and for images with transferred color (on the right).

Figure 16: Results of Color Transfer between high resolution images ($\approx 10$ megapixel) by a pixel-wise cycle monotone mapping.

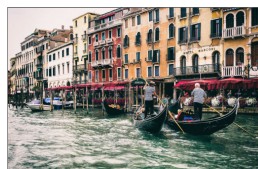 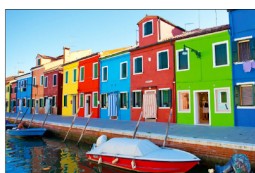 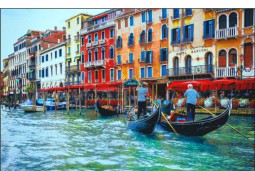 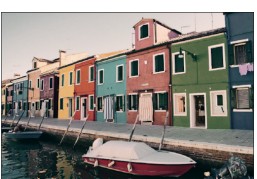

(a) Original images (on the left) and images obtained by color transfer (on the right).

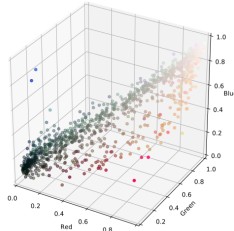 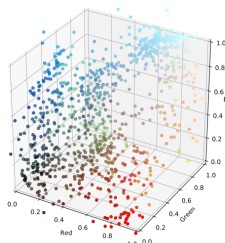 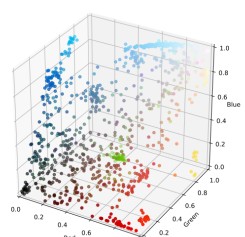 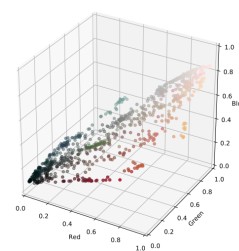

(b) Color palettes (3000 random pixels, best viewed in color) for the original images (on the left) and for images with transferred color (on the right).

Figure 17: Results of Color Transfer between images by a pixel-wise cycle monotone mapping.

(2016; 2017); Redko et al. (2018), while some approaches adopt neural networks to estimate the mapping $g$ Bhushan Damodaran et al. (2018); Seguy et al. (2017).

We address the unsupervised domain adaptation problem which is the most difficult variant of this task. Labels are available only in the source domain, so we do not use any information about the

labels. Our method trains $g$ as a gradient of a convex function. It can be applied to new arriving samples which are not present in the train set.

We test our model on MNIST ($\approx 60000$ images; $28 \times 28$) and USPS ($\approx 10000$ images; rescaled to $28 \times 26$) digits datasets. We perform **USPS $\rightarrow$ MNIST** domain adaptation. To do this, we train LeNet $\geq 99\%$-accuracy classifier $h$ on MNIST. Then, we apply $h$ to both datasets, extract 84 last layer features. Thus, we form distributions $\mathbb{Q}$ (features for MNIST) and $\mathbb{P}$ (features for USPS).

To fit a cycle monotone domain adaptation mapping, we use DenseICNN [32; 128; 128, 128] potentials. We train our model on mini-batches of 64 samples for 10000 iterations with cycle regularization $\lambda = 1000$. We use Adam optimizer with lr $= 10^{-4}$ and impose $10^{-7}$ $\mathcal{L}^1$-penalty on the weights of the networks.

Similar to Seguy et al. (2017), we compare the accuracy of MNIST 1-NN classifier $f$ applied to features $x \sim \mathbb{P}$ of USPS with the same classifier applied to mapped features $g^{\dagger}(x)$. 1-NN is chosen as the classification model in order to eliminate any influence of the base classification model on the domain adaptation and directly estimate the effect provided by our cycle monotone map.

The results of the experiment are presented in Table 3. Since domain adaptation quality highly depends on the quality of the extracted features, we repeat the experiment 3 times, i.e. we train 3 LeNet MNIST classifiers for feature extraction. We report the results with mean and central tendency. For benchmarking purposes, we also add the score of 1-NN classifier applied to the features of USPS transported to MNIST features by the discrete optimal transport. It can be considered as the "most straightforward" optimal transport map.[15]

|  | Repeat 1 | Repeat 2 | Repeat 3 | Average ($\mu \pm \sigma$) |
|---|---|---|---|---|
| Target features | 75.7% | 77% | 75.4% | $76 \pm 0.8\%$ |
| **Mapped features (W2GN)** | 80.6% | 80.3% | 82.7% | $81.2 \pm 1\%$ |
| Mapped features (Discrete OT) | 76% | 75.7% | 76.1% | $75.9 \pm 0.4\%$ |
| Mapped features Seguy et al. (2017) | - | - | - | 77.92% |

Table 3: 1-NN classification accuracy on USPS $\rightarrow$ MNIST domain adaptation problem.

Our reported scores are comparable to the ones reported by Seguy et al. (2017). We did not reproduce their experiments since Seguy et al. (2017) does not provide the source code for domain adaptation. Thus, we refer the reader directly to the paper's reported scores (Table 1 of Seguy et al. (2017), **first** column with scores).

For visualization purposes, we plot the two main components of the PCA decomposition of feature spaces (for one of the conducted experiments) in Figure 18: MNIST features, mapped USPS features by using our method, original USPS features.

## C.8   IMAGE-TO-IMAGE STYLE TRANSFER

We experiment with ConvICNN potentials on publicly available[16] **Winter2Summer** and **Photo2Cezanne** datasets containing $256 \times 256$ pixel images.

We train our model on mini batches of 8 randomly cropped $128 \times 128$ pixel RGB image parts. As an additional augmentation, we use random rotations ($\pm \frac{\pi}{18}$), random horizontal flips and the addition of small Gaussian noise ($\sigma = 0.01$). The networks are trained for 20000 iterations with cycle regularization $\lambda = 35000$. We use Adam optimizer and impose additional $10^{-1}$ $\mathcal{L}^1$-penalty on the weights of the networks. Our scheme of style transfer between datasets is presented in Figure 19. We provide the additional results for Winter2Summer and Photo2Monet datasets in Figures 20b and 20a respectively.

In all the cases, our networks change colors but preserve the structure of the image. In none of the results did we note that the model removes large snow masses (for winter-to-summer transform) or

---

[15]In contrast to our method, it can not be directly applied to out-of-train-sample examples. Moreover, its computation is infeasible for large datasets.

[16]https://github.com/junyanz/pytorch-CycleGAN-and-pix2pix

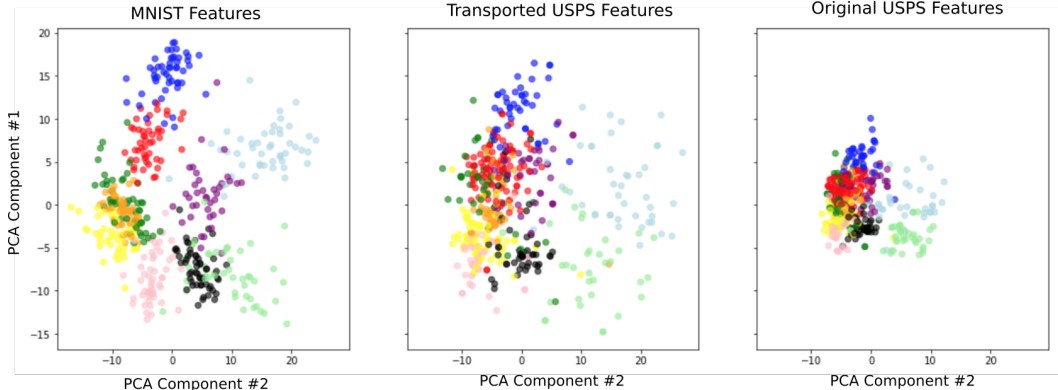

Figure 18: A pair of main principal components of feature spaces. From left to right: MNIST feature space, mapped USPS features by W2GN, original USPS feature space. PCA decomposition is fitted on MNIST features. Best viewed in color (different colors represent different classes of digits $0 - 9$).

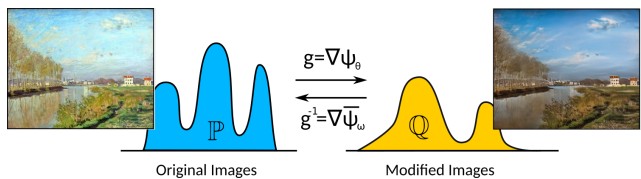

Figure 19: Schematically presented image-to-image style transfer by a pair of ConvICNN fitted by our method.

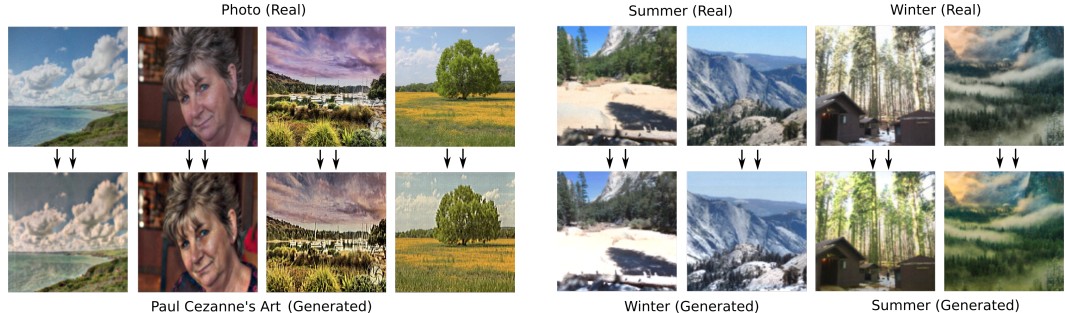

(a) Results of cycle monotone image-to-image style transfer by ConvICNN on Photo2Cezanne dataset, $128 \times 128$ pixel images.

(b) Results of cycle monotone image-to-image style transfer by ConvICNN on Winter2Summer dataset, $128 \times 128$ pixel images.

Figure 20: Additional results of image-to-image style transfer on Winter2Summer and Photo2Monet datasets.

covers green trees with white snow (for summer-to-winter). We do not know the exact explanation for this issue but we suppose that the desired image manipulation simply may not be cycle monotone.

For completeness of the exposition, we provide some results of cases when our model does not perform well (Figure 21). In Figure 21a the model simply increases the green color component, while in Figure 21b it decreases this component. Although in many cases it is actually enough to transform winter to summer (or vice-versa), sometimes more advanced manipulations are required.

In the described experiments, we applied our method directly to original images without any specific preprocessing or feature extraction. The model captures some of the required attributes to transfer, but sometimes it does not produce expected results. To fix this issue, one may consider OT for the quadratic cost defined on **features** extracted from the image or on **embeddings** of images (similar to

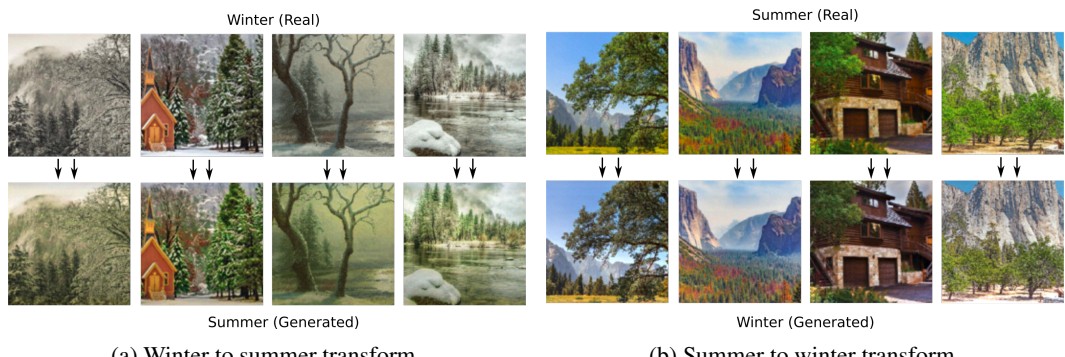

(a) Winter to summer transform.      (b) Summer to winter transform.

Figure 21: Cases when the cycle monotone image-to-image style transfer by ConvICNN on Winter2Summer dataset works not well, $128 \times 128$ pixel images.

the domain adaptation in Subsection C.7 or latent space mass transport 5.2). This statement serves as the challenge for our further research.

