# OpenReview forum: "Wasserstein-2 Generative Networks"
_ICLR.cc/2021/Conference — ICLR 2021 Poster_

### Official Review · AnonReviewer1 · 2020-10-29

**Rating:** 5
**Confidence:** 4

**Review:**

This paper proposes Wasserstein-2 Generative Networks (W2GNs) which is an optimal transport framework for learning generative models. Unlike minimax problems of Wasserstein GANs, the proposed approach which is based on minimizing the 2-Wasserstein distance reduces to a single-level optimization problem. The paper numerically shows that the new approach enjoys faster convergence and improves upon the performance scores of Wasserstein GANs and other optimal transport baselines. While the paper's idea on applying optimal transport tools for training generative models seems interesting, the discussed theoretical and numerical results are not supportive enough to show that the proposed approach indeed improves upon WGANs. Also, the theoretical sections have been written in a convoluted way with several weakly supported claims and the final algorithm has not been stated clearly. I, therefore, do not recommend the paper for acceptance.

To further explain my concerns, let me start with section 3 which reviews the dual formulation to 2-Wasserstein distance in Eq. (8) and also the connection to the convex conjugate optimization in Eq. (9). Here, the paper vaguely mentions that the optimization problem for computing the convex conjugate is "convex and very complex" followed by a brief explanation which I do not find satisfactory. Specifically, the paper's way of reasoning does not convince me why it is necessary to switch from Eq. (8)  to the paper's formulation in (12). Both the issues mentioned for problems (8) and (9) also apply to the formulation in (12) as (12) still requires taking the gradient of the convex conjugate \psi_w. Also, I highly recommend replacing the terms "very complex" and "impossible" with more solid theoretically or numerically supported statements.

Next, let me refer to the final sentences of the first paragraph of section 4.1: " Yet such a problem is still minimax. Thus, it suffers from typical problems such as convergence to local saddle points, instabilities during training and usually requires non-trivial hyperparameters choice." These sentences argue that every minimax problem suffers from instability and convergence to local solutions. However, the paper's own formulation in (12) also leads to a non-convex optimization problem for which the authors show no convergence guarantees to a global solution. The paper should either remove these sentences or precisely explain why the proposed non-convex problem enjoys better convergence properties than the minimax problems. Let me also add that while Eq. (12) states an optimization problem, it still does not completely characterize a learning algorithm, because it is unclear how one wants to take the gradient of \psi_w's convex conjugate. I think the paper should include an algorithm clearly stating the steps of learning the generative model.

Finally, the theoretical guarantees in Theorem 4.1 and 4.2 do not analyze the algorithm's performance for the class of input convex neural nets. Theorem 4.1 connects the optimization error to the closeness of the generative and underlying distributions. Yet, it does not provide any guarantee on how large the optimization error could be for convex neural nets. The result of Theorem 4.2 also immediately follows from the assumptions and offers little understanding of the algorithm's performance with convex networks, since it considers the set of all differentiable functions instead. The theoretical guarantees should somehow analyze the algorithms' convergence and approximation properties for convex neural nets rather than all differentiable convex functions. Overall, the paper seems to carry several nice ideas, but the theoretical discussion needs to be significantly improved.

*****
Review update: I thank the authors for their response and for revising the paper based on the comments. The revision addresses several of my concerns. I still think the theoretical guarantees should be stronger and therefore change my score to borderline 5.

---

> ### Author Response · Authors · 2020-11-15
> **Answers to AnonReviewer1**
>
> Thank you for your valuable feedback. Please find above (in our reply to all the Reviewers) the answers to your questions common with other reviews. Please find below our answers to your questions that do not overlap with those of other Reviewers.
>
> **[1] Conjugacy of neural networks**
>
> First we note, that in our method, analogously to Makkuva et. al. (2020), the optimal discriminators $\psi^{\star}$ and $\overline{\psi^{\star}}$ are approximated by two separate neural networks $\psi_{\theta}$ and $\overline{\psi_{\omega}}$ respectively. In particular, $\overline{\psi_{\omega}}$ is itself a neural network (not a convex conjugate function to some neural network $\psi_{\omega}$).  The approximate forward and inverse transport maps are the gradients of these networks, i.e. $\nabla\psi_{\theta}$ and $\nabla\overline{\psi_{\omega}}$. During optimization, there is no need to compute the conjugate functions of $\psi_{\theta}$, $\overline{\psi_{\omega}}$ or the inverse functions for $\nabla\psi_{\theta}$, $\nabla\overline{\psi_{\omega}}$. Thus, in constrast to Makkuva et.al. (2020) and Taghvaei et.al. (2019), in our method there are no additional optimization subproblems.
>
> **[2] Why non-minimax is better than minimax**
>
> In the experimental section, we compare a feasible property of non-minimax vs. minimax optimization, i.e. convergence speed. We demonstrate that our non-minimax method for computing W2 optimal transport maps converges much faster than its closest alternative, i.e. minimax approach by Makkuva et. al. (2020). In Figure 2 (Section 5.1) and Figure 2 (Appendix C.3), we show that our non-minimax setup converges up to 10x faster in high dimensions. As we explain in Section 5.1, this naturally follows from the fact that our optimization does not require solving inner optimization subproblem.
>
> You also noted that it is unclear why one should use our method (Eq. (12)) instead of the minimax method by Taghvaei et.al. (2019) (represented by Eq. (8) in our paper). For brevity, let us name this minimax method by MM-1 and its successor method of Makkuva et.al. (2020) [discussed in the previous paragraph] by MM-2. MM-1 method uses only one ICNN (discriminator) and solves an additional convex subproblem to compute the value of the conjugate discriminator. We initially did not include any comparison with the MM-1 method, since Makkuva et.al. (2020) [Remark 3.5] has already explained why their MM-2 method (which we outperform) is superior to MM-1 method.
>
> Nevertheless, we are planning to add MM-1 method to our comparison in Section 5.1 in the next revision to demonstrate that MM-1 performs even worse than MM-2 w.r.t. the computational time.
>
> Please tell us whether this comparison is needed and will help to convince you that MM-1 method has poor performance (in terms of the computational time).
>
> **[3] Theoretical Results**
>
> In this paper, we do not theoretically analyze the optimization error (or provide convergence guarantees), but analyze approximation error assuming that there is no optimization error.
>
> We demonstrate (Theorem 4.2) that if the approximating class of convex functions (e.g. ICNNs) is rich enough, then the minimizer of eq. (12) is a good approximator of the OT map. On the other hand, the approximation class can actually be made rich enough by considering a large ICNN because fully connected ICNNs satisfy the universal approximation property. The question how large ICNN architecture should be to provide a required approximation is a complicated question which depends on the many additional aspects: the parameterization, the complexity of the pair of distributions, etc. Answering this question in the context of W2 transport by ICNNs  might be the avenue for a future research.
>
> Please note that we analyse the performance of our method in some arbitrary given classes $\Psi_X$ and $\overline{\Psi_Y}$ of differentiable convex functions, e.g. represented by ICNNs. The ICNNs which we use are actually differentiable, since we use the CELU activation function instead of ReLU (all technical details on the implementation are given in Appendix B).

---

### Official Review · AnonReviewer2 · 2020-10-29
**An end-to-end algorithm with a non-minimax objective for training a OT map for quadratic cost**

**Rating:** 8
**Confidence:** 4

**Review:**

From my perspective, this is a much needed and love-to-see work for the line of neural generative modeling. Previous approaches were dominated by GAN based approaches which require solving a minimax optimization which has technical hurdles in practice. This paper reviews the literature of OT theory, and provides a comprehensive explanation about the relation between W_2 and cyclical monotonic map. In particular, it motivates the opportunity to approximate Eq. (8) with Eq.(12), as which a non-minimax formulation exist.

Pros:

- a non-minimax formulation was proposed for neural generative modeling
- theoretical properties are derived for the proposed formulation Eq.(12)
- experiments are preliminary but promising.

Cons:

- The paper misses a discussion about how gradient-based optimization is done for Eq.(12) in implementation. For example, it is not clear what functions are supposed to be parameterized as a neural network. Is it \varphi_\theta and \bar{\varphi}_\omega? If so, how the gradient is calculated for \Delta \bar{\varphi_\omega}? Would it involve high order auto-differentiation?
- The authors have not outlined how computational feasible their approach is? This makes me less confident about their approach regarding processing more difficult datasets or tasks.
- I strongly suggest the authors to re-organize their presentations, and focus more on what was actually calculated in practice. This allows others to follow their work and reproducing their experiments. Some mathematical introduction is nice, but they have to be directly related to the approach of this paper.



---------------
post revision:
I read authors' revision, and it is indeed an improved version with more readabilities. Since I am familiar with the OT literature, the mathematical presentation is clear enough for me to follow. The motivation of this paper is clear. The presentation of algorithm is also clear enough in the revision. But I can not say much for wider audience this paper may target.

Given the significance and popularity of GAN related work, this paper seems a big deal to the community. This's why I intend to give 8 instead of 7 for the revised paper.

---

> ### Author Response · Authors · 2020-11-15
> **Answers to AnonReviewer2**
>
> Thank you for your valuable feedback. Please find above (in our reply to all the Reviewers) the answers to your questions common with other reviews. Please below find our answer to your question not which is not common with the other reviews.
>
> **Q: Would it [the optimization procedure] involve high order auto-differentiation?**
>
> As we noted in the general answer to all the reviewers, the computational time is linear w.r.t. the computation time for a single forward pass for $\psi_{\theta}(x)$. While it might seem that the full second order derivative $\frac{\partial^{2}}{\partial (\theta,\omega)\partial (x,y)}$ (a part of the Hessian H of objective w.r.t. $(\theta,\omega,x,y)$) is needed, actually only its product Hv with some direction v is used (known as second directional derivative). Its computation is as fast as computation of the gradient, see http://citeseerx.ist.psu.edu/viewdoc/summary?doi=10.1.1.29.6143

---

> > ### Comment · AnonReviewer2 · 2020-11-17
> > **Look forward to your updated version.**
> >
> > I will calibrate my score based on the revision.

---

> > > ### Author Response · Authors · 2020-11-22
> > > **Thank you for considering the updated paper**
> > >
> > > Dear reviewer, thank you very much for your valuable comments and feedback. We really appreciate that you even raised the grade and are very positive about our results.

---

### Official Review · AnonReviewer5 · 2020-11-05
**Reasonable approach but writing to be improved**

**Rating:** 6
**Confidence:** 2

**Review:**

The paper proposes a new method for learning an optimal pushforward (for the quadratic cost) from a distribution to another distribution based on samples of both distributions. The optimal transport problem is first written equivalently as a minimax problem over set of convex functions, as in Makkuva et al. 19. Then, the convex functions are parametrized as Input Convex Neural Networks (ICNN). To my understanding, the novelty of the approach is to replace the minimax problem over the parameters of the ICNN by an easier to solve a regularized minimization problem. The authors prove the consistency of their approach: loosely speaking, they show that true minimizers of their approximate problem are epsilon minimizers of the original optimal transport problem. They also provide promising numerical experiments.



The paper proposes a simple fix to drawbacks of previous works of on the topic, which e.g. propose minimax approaches. Moreover, the consistency of the approach is rigorously proven.
On the other hand, one could argue that the novelty is marginal (replacing a minimax problem by a regularized minimization). Moreover, the consistency result assumes access to a true minimizer of the regularized problem (which is highly non convex). Finally, the writing of the paper can be improved.


I rate the paper as marginally below.


Regarding the novelty of the approach. This can be addressed by SOTA numerical experiments, but this is not the case (although the numerical experiments are well explained and detailed).
Regarding the consistency result. The authors provide an approximation result that does not take into account the optimization of the regularized problem. It is a difficult a problem but the authors could at least mention this difficulty.
Regarding the overall quality of the paper.
- The structure of the introduction is strange. Several paragraphs are used to explain points which cannot be understood at this stage or provide too much details. Several paragraphs are used to explain that we look for well structured pushforward mappings. Why not directly mention the equivalent optimal transport problem ?
In dimension one all continuous invertible mappings sending P to Q are monotone? I am not sure. What is a maximal monotone mapping (eq 2)? Considerations about the dimension of the space should not be in the intro in my opinion (the reader has the feeling that the authors are defending themselves). Considerations on minimax vs non minimax and end to end are too difficult to understand at this stage (same in Section 2).
-Section 2. The notation for the Fenchel transform is misleading. It should be with a star...
-Section 3. Why do we need a positive density? Optimal pushforward mappings exist (in both directions) if both distributions admit a density wrt Lebesgue measure. What is the point of the footnote 4? The set "Convex" is not defined. Convex potentials are called discriminators. Why this name? I guess that there is a connection with GANs but I don't see (actually this work is about fitting an optimal pushforward mapping, which is something independent from the GAN setup).
-Section 4. Check Eq 10 (symbol = and alignment). Consideration on why non minimax is better than minimax should be explained before.  "Fully-connected ICNNs satisfy universal approximation property". What does it imply for the assumptions of Th 4.2? Are they satisfied or not?



MINOR:
- Question: Are gradients wrt to x of the ICNN easy to compute?
- parenthesis in the equation after Forward Generative Property
- "diffirentiable" (several times)
- I have seen similar considerations (using ICNN to fit gradient of convex functions) in https://openreview.net/pdf?id=rklx-gSYPS

---

> ### Author Response · Authors · 2020-11-15
> **Answers to AnonReviewer5**
>
> Thank you for your valuable feedback. Please find above (in our reply to all the Reviewers) the answers to your questions common with other reviews. Please below find below our answers to your questions that do not overlap with those of other Reviewers.
>
> **[1] Every 1D continuous and invertible function is monotone**
>
> This is true, please check the discussion here
> https://math.stackexchange.com/questions/2476832/continuous-function-is-invertible-only-if-it-is-strictly-monotonic
>
> **[2] The notation of Fenchel transform**
>
> We use overline notation rather that the superscript (*) to avoid the notation with multiple superscripts (e.g. dagger superscripts are already used to denote the approximating functions). If you can kindly suggest some other notation, please recommend.
>
> **[3] Positive Density**
>
> You noted that it is not needed to require positive density everywhere. We completely agree --- in all our theoretical results this requirement can be removed.
>
> **[4] Potentials/Discriminators Notation**
>
> We call functions $\psi$ as discriminators in order to emphasize that they play the role of the critic between a pair of distributions. Following you advice, we will replace the "discriminator" notation by the "potential" one.
>
> **[5] The Universal Approximation Property and Theorem 4.2**
>
> Theorem 4.2 demonstrates that if the approximating classes of convex functions $\Psi_X$, $\overline{\Psi_Y}$ (e.g. ICNNs) contain transport maps that are $\epsilon_{X}$-- and $\epsilon_{Y}$--close to the true ones, then as the result of the optimization (assuming that we achieve the optimum within those classes), we obtain $(\psi^{\dagger},\overline{\psi^{\ddagger}})$ for which the value of the objective is $O(\epsilon_{X}+\epsilon_{Y})$-optimum. Next, by using our Theorem 4.1, we immediately obtain that transport maps $\nabla\psi^{\dagger}$, $\nabla\overline{\psi^{\ddagger}}$ are $O(\epsilon_{X}+\epsilon_{Y})$-close to the optimal forward $\nabla\psi^{\star}$ and inverse $\nabla\overline{\psi^{\star}}$  maps respectively in $W_{2}^2$ sense. The approximation classes $\Psi_X$, $\overline{\Psi_Y}$ can be made rich enough ($\epsilon_{X}, \epsilon_{Y}\rightarrow 0$) by considering a large ICNN since fully connected ICNNs satisfy the universal approximation property. Thus, we can fit the transport map with any desired error (see the end of Appendix A.2 for additional details). Note that this is not the case with entropy-based regularization of Seguy et. al. (2017) since their solution will always be biased from the true one.
>
> We also emphasize that for the min-max method there is no analogous theoretical result. Theorem 3.5 of Makkuva et.al. (2020) provides an upper bound on the closeness of the fitted forward transport map $\nabla\psi^{\dagger}$ to the true $\nabla\psi^{\star}$. The upper bound is linear in the sum of the maximization gap and the minimization gap of their min-max problem. However, it remains unclear whether the optimal solution of min-max problem within some restricted classes $\Psi_X$, $\overline{\Psi_Y}$ actually provides small values of the sum of mentioned gaps, even when $\Psi_X$, $\overline{\Psi_Y}$ are rich enough.

---

> > ### Comment · AnonReviewer5 · 2020-11-23
> > **Thanks**
> >
> > OK, I thank the authors for their reply and I am raising my score based on their answers.
> > I encourage the authors to mention that Section 4.2 does not take into account the optimization properties.

---

> > > ### Author Response · Authors · 2020-11-23
> > > **Thank you for considering the updated paper**
> > >
> > > Dear reviewer, thank you very much for your valuable comments and feedback. We really appreciate that you even raised the grade and you are positive about our results. Following your advice, we will additionally mention in the paper that Section 4.2 (in particular, Theorem 4.2) does not take into account the optimization error.

---

### Author Response · Authors · 2020-11-15
**Answers to shared questions**

Dear reviewers, thanks for your insightful comments! We are currently working on improving the paper according to your comments and will soon submit the updated version. Please, find the answers to your shared questions below.

**[1] Details of the Numerical Optimization Procedure: Reviewers 1 and 2**

 To address this concern, we are planning to add a step-by-step detailed description of the numerical procedure used to optimize our main objective Eq. (12). Please note that the architecture, the implementation and the training details are  described in Appendices B, C of the initial submission.

**[2] Computational Complexity: Reviewers 1, 2 and 5**

As we note in Appendix C.1, to compute the gradients of the objective w.r.t. parameters of the ICNNs (for SGD steps, we use the automatic differentiation provided by the PyTorch framework. We emphasize that computing the forward and the backward passes of Eq. (12) is NOT difficult, see below.

The time complexity is comparable (up to a constant factor) to that of a single forward pass through the discriminator $\psi_{\theta}(x)$.  This claim follows from the well-known fact that gradient evaluation $\nabla_\theta h_\theta(x)$ of $h_\theta: \mathbb{R}^{D} \to \mathbb{R}$, when parameterized as a neural network, requires time proportional to the size of the computational graph. Hence gradient computation requires computational time proportional to the time for evaluating the function $h_\theta(x)$ itself. The same holds true when computing the derivative with respect to $x$. Thus, the number of operations required to compute different terms in Eq. (12), e.g. $\nabla \overline{\psi_{\omega}}(y)$,  $\psi_{\theta}\big(\nabla \overline{\psi_{\omega}}(y)\big)$ and $\nabla\psi_{\theta}\circ\nabla \overline{\psi_{\omega}}(y)$, is also linear w.r.t. the computation time of $\psi_{\theta}(x)$ or, equivalently, $\overline{\psi_{\omega}}(x)$. As a consequence, the time required for the forward pass of Eq. (12) is larger than the forward pass for $\psi_{\theta}(x)$ only up to a constant factor. Thus, the backward pass for Eq. (12) with respect to parameters of ICNNs $\theta$ and $\omega$ is also linear in the computation time of $\psi_{\theta}(x)$. We empirically measured that for our DenseICNN discriminators, the computation of gradient of Eq. (12) w.r.t. parameters $\theta,\omega$ requires roughly 8-12x more time than the computation of $\psi_{\theta}(x)$.

Please note that for our model in the initial submission, we provide the convergence time in the experiments with the Gaussian setting. In Figure 2, we demonstrate that the method converges in less than 10 minutes for the highest considered dimension 4096. In Figure 12 of Appendix, we also provide wall clock times for smaller dimensions. The wall clock times for other experiments are as follows (not reported in the initial submission): $<2$ minutes for color transfer and domain adaptation, several hours (roughly $<4$) for latent space mass transport and style transfer.

---

> ### Author Response · Authors · 2020-11-17
> **Rebuttal Revision**
>
> Dear reviewers, we have revised the paper according to your comments. Please consider the updated submission.
>
> The edits are highlighted by the blue color in the revised version of the paper. The main edits are listed below.
>
> **(R1, R2)** We added the detailed numerical optimization algorithm to Section 4.1.
>
> **(R1, R2, R5)** We devoted Section C.2 to the discussion of the computation complexity of our training procedure.
>
> **(R1)** We added the comparison with the minimax approach by Taghvaei and Jalali (2019) to Section 5.1 (Table 1, Figure 2) and Appendix C.4 (Figure 12).
>
> **(R1)** In several relevant places of the paper (paragraphs with contributions in Introduction, the algorithm Section 4.1, etc.), we additionally highlighted the key feasible advantage w.r.t. minimax optimization, i.e. faster convergence.
>
> **(R5)** We changed the "discriminators" notation to "potentials" notation.
>
> **(R5)** We removed the (unnecessary) requirement of positive density from the theorems.
>
> If there are any additional changes you suppose we should perform, please kindly suggest.

---

### Decision · Program_Chairs · 2021-01-07
**Final Decision**

**Decision:**

Accept (Poster)

**Comment:**


The reviewers have  different views on the papers but agreed that the paper can be accepted. However, they suggested
some points of improvements including the writing (clarity and style) and experiments showing strong improvements
compared to WGAN.